# Global diversity and biogeography of potential phytopathogenic fungi in a changing world

Pengfa Li [1,2,11], Leho Tedersoo [3,11], Thomas W. Crowther [4,11], Baozhan Wang [1] ✉, Yu Shi[5], Lu Kuang [1], Ting Li[1], Meng Wu[2], Ming Liu[2], Lu Luan [2], Jia Liu[6], Dongzhen Li[7], Yongxia Li[7], Songhan Wang[8], Muhammad Saleem[9], Alex J. Dumbrell [10] ✉, Zhongpei Li[2] & Jiandong Jiang [1] ✉

Phytopathogenic fungi threaten global food security but the ecological drivers of their global diversity and biogeography remain unknown. Here, we construct and analyse a global atlas of potential phytopathogenic fungi from 20,312 samples across all continents and major oceanic island regions, eleven land cover types, and twelve habitat types. We show a peak in the diversity of phytopathogenic fungi in mid-latitude regions, in contrast to the latitudinal diversity gradients observed in aboveground organisms. Our study identifies climate as an important driver of the global distribution of phytopathogenic fungi, and our models suggest that their diversity and invasion potential will increase globally by 2100. Importantly, phytopathogen diversity will increase largely in forest (37.27-79.12%) and cropland (34.93-82.51%) ecosystems, and this becomes more pronounced under fossil-fuelled industry dependent future scenarios. Thus, we recommend improved biomonitoring in forests and croplands, and optimised sustainable development approaches to reduce potential threats from phytopathogenic fungi.

Phytopathogenic fungi pose a major threat to global food security, ecosystem service delivery and human livelihoods[1,2]. Plant diseases have a range of direct, quantifiable economic consequences for crop and forest management across a range of terrestrial environments[3,4]. For example, the persistent presence of these diseases would only leave enough food to feed ~1/3 of the world's population, and a handful of plant diseases in forests could reduce global $CO_2$ absorption annually by 230–580 megatons[5]. Attempts to mitigate the effects of such diseases caused by phytopathogenic fungi, via chemical fungicides and biocontrol agents, are often locally ineffective and globally inadequate due to the complexity and diversity of phytopathogenic fungal communities and an incomplete understanding of the factors regulating them[6,7]. Therefore, ascertaining the distribution and the environmental attributes that structure phytopathogenic fungal communities across the globe was recently considered to be a priority research direction[8].

[1]Department of Microbiology, College of Life Sciences, Nanjing Agricultural University, Key Laboratory of Agricultural and Environmental Microbiology, Ministry of Agriculture and Rural Affairs, 210095 Nanjing, China. [2]State Key Laboratory of Soil and Sustainable Agriculture, Institute of Soil Science, Chinese Academy of Sciences, 210008 Nanjing, China. [3]Mycology and Microbiology Center, University of Tartu, Tartu, Estonia. [4]Institute of Integrative Biology, ETH Zürich, 8092 Zürich, Switzerland. [5]State Key Laboratory of Crop Stress Adaptation and Improvement, School of Life Sciences, Henan University, Kaifeng, China. [6]Soil and Fertilizer & Resources and Environment Institute, Jiangxi Academy of Agricultural Sciences, 330200 Nanchang, China. [7]Key Laboratory of Forest Protection of National Forestry and Grassland Administration, Ecology and Nature Conservation Institute, Chinese Academy of Forestry, 100091 Beijing, China. [8]College of Agriculture, Nanjing Agricultural University, 210095 Nanjing, China. [9]Department of Biological Sciences, Alabama State University, Montgomery, AL 36104, USA. [10]School of Life Sciences, University of Essex, Colchester, Essex, UK. [11]These authors contributed equally: Pengfa Li, Leho Tedersoo, Thomas W. Crowther. ✉e-mail: bzwang@njau.edu.cn; adumb@essex.ac.uk; jiang_jjd@njau.edu.cn

The study of microbial biogeography enables researchers to link microbial communities to macroecology through revealing where microorganisms live, at what abundance, and why[9]. To date, this approach has successfully described and analysed the global atlases of multiple microbial organisms, including bacteria[10–12], total fungi[1,13,14] and protists[15,16]. However, despite their ecological and agricultural importance, very few studies have focused on phytopathogenic fungal distributions at the global scale, but with limited taxonomic and environmental coverage[17], restricting our understanding of the role of climate in shaping these communities into the future. Given the tightly coupled relationships between climate and pathogen development[18], climate change has increased the prevalence and severity of some human, animal and plant diseases[19]. While predicting the consequences of climate change remains challenging, models have already predicted that crop yields[20], carbon sequestration[21] and pollination rates[22] are expected to decrease, whilst evapotranspiration and tree mortality are expected to increase under future climate scenarios[23]. In addition, as the geographic ranges of plant species shift in response to climate change and alien species are introduced, the potential of emerging novel plant pests and pathogens increases[17,22], leading to increased incidence and severity of the diseases they cause[24]. Considering the magnitude of global climate change, it is imperative to determine how a changing climate affects the distribution of phytopathogenic fungi, and potentially to use this new knowledge to inform policies to control the emergence of future plant diseases and maintain ecosystem functions and services.

Here, we use a global dataset that combines newly generated DNA sequence data with previously published mycobiome sequences from the GlobalFungi database[25]. In total, our global dataset included 5753 potential phytopathogenic species hypotheses (hereafter ppSHs; ppSHs were generated by 98.5% ITS sequence similarity, which can be treated as potential phytopathogenic fungal species) from 20,312 samples distributed across all seven continents and four major oceanic island regions, 11 land cover types (forests, grasslands, croplands, aquatic, deserts, woodlands, shrublands, tundra, wetlands, urban and mangroves) and 12 habitat types (soils, plant shoots, roots, rhizosphere, deadwood, air, sediment, litter, lichen, freshwater, topsoil and dust (atmospheric deposition)) (Fig. 1). We used this to comprehensively describe the global biogeographic patterns of potential phytopathogenic fungi (hereafter phytopathogenic fungi for conciseness), and to identify the ecological drivers regulating their diversity. Through a series of theoretical and modelling approaches, we (i) mapped the distribution of phytopathogenic fungi and revealed the underlying factors regulating their distribution, (ii) uncovered universal ecological properties of phytopathogenic fungal communities and their dynamics, and (iii) predicted changes in their diversity and invasion potential under future climate change scenarios. While our study uncovers the global patterns of phytopathogenic fungi, many of the proposed relationships between climate, phytopathogen diversity and disease severity will need to be addressed using specifically targeted experimental approaches elsewhere.

## Results and discussion
### Global distribution of potential phytopathogenic fungi
Globally, ppSHs represented between 0% and 99.64% (mean = 8.96%) of all fungal ITS sequences per sample (Supplementary Fig. 1a). The relative abundance of ppSHs was highest in the Indian Ocean (19.17%), North America (11.51%), and Europe (11.25%) regions; tundra (19.00%), urban (17.60%), and mangrove (14.32%) land cover types; and in air (40.37%), dust (24.16%) and plant shoot (20.42%) habitats (Supplementary Fig. 1b). Moreover, the relative abundance of ppSHs in the soil habitat of our main dataset (4.24%) can be cross-validated by a recently published Global Soil Mycobiome consortium (GSMc) dataset[14] (relative abundance of ppSHs: 4.78%; Supplementary Fig. 1b). The most abundant phytopathogenic fungal genera included *Fusarium,*

*Alternaria, Fusicladium, Neoerysiphe* and *Mycosphaerella*, whilst the most frequently occurring phytopathogenic fungal genera included *Fusarium, Trichoderma, Alternaria, Epicoccum* and *Mycosphaerella* across all sampling sites (Fig. 2a), regions (Supplementary Data 1), land cover types (Supplementary Data 2) and habitat types (Supplementary Data 3). Given that some phytopathogens contribute additional ecological roles[26–28], we divided the ppSHs into two groups based on their trophic modes, namely exclusive modes (plant pathogens only, 2405 ppSHs, 41.8% of all pathogenic phylotypes) and non-exclusive modes (plant pathogen and endophyte and/or saprotrophic fungi, 3348 ppSHs, 58.2% of all pathogenic phylotypes). We then mapped the global distribution of phytopathogenic fungi using simple linear regression (Supplementary Fig. 2b), second-order polynomial regression (Fig. 2b) and a Generalised Linear Model (GLM; Fig. 2c); our results consistently indicated a relatively weak relationship between latitude and the relative abundance of phytopathogenic fungi (for phytopathogens with both exclusive and non-exclusive trophic modes, Supplementary Fig. 2c, d). However, the phytopathogens with exclusively phytopathogenic trophic modes ($R^2 = 0.005$, $F = 46.6$, Supplementary Fig. 2c) showed a marginally weaker relationship between their relative abundances and latitude than those with non-exclusive trophic modes ($R^2 = 0.006$, $F = 63.6$, Supplementary Fig. 2d).

### Global patterns of phytopathogenic fungal richness
Across all samples, the richness of phytopathogenic fungi (number of observed ppSHs) ranged from 1 to 372 (mean = 28.07) ppSHs (Supplementary Fig. 1c). The ppSH richness was highest in Asia (40.80), Africa (35.46) and Europe (28.96), and it was particularly high in urban (64.54), cropland (44.59), and grassland (35.05) land cover types (Supplementary Fig. 1d). Topsoils had extremely high ppSH richness (68.56), highlighting the importance of these habitats as reservoirs of fungal plant pathogens in natural ecosystems[6].

The relationship between ppSH richness and latitude was evaluated using simple linear regression and second-order polynomial regression for this dataset (Fig. 2d and Supplementary Fig. 3a) and the GSMc dataset[14]. The second-order polynomial best described trends in our data and had a higher fitting efficiency based on multiple parameters than a linear or higher-order polynomial fits ($R^2$, $F$, AIC and $P$ value). In addition, we inferred the global pattern of ppSH richness using a GLM after cross-validation (Fig. 2e and Supplementary Fig. 3b). Both second-order polynomial regression and the GLM consistently demonstrated that ppSH richness peaks at intermediate latitudes, with lower values in equatorial and polar regions (Fig. 2d, e). Such a richness-latitude relationship was also evident across the main land cover types and habitats (Supplementary Fig. 3c). The soil-only GSMc dataset of mostly forest habitats showed higher ppSH richness at both low and intermediate latitudes (Supplementary Fig. 3d). Our results provide evidence that the traditional latitudinal diversity gradient (LDG) is poorly applicable for ppSHs. This contrasts with the traditional LDG of plants, arthropods, vertebrates[29] and some bacterial groups[10]. Moreover, the distribution of ppSH diversity is also greatly different from that of total fungi and arbuscular mycorrhizal fungi, whose diversity was demonstrated to peak in tropical areas by multiple previous studies[1,30–32]. This suggests that fungi with pathotrophic modes may have distinct biogeographical patterns compared to other fungi with different trophic modes (e.g., saprotroph and symbiotroph), as a consequence of differential life-history strategies and community assembly mechanisms[33]. After regrouping the samples according to sequencing region (ITS1, ITS2 and both), the ppSH richness also consistently peaks at intermediate latitudes (Supplementary Fig. 3e). In addition, we tested the richness-latitude relationship using a rarefied dataset (4000 reads per sample), and the results also indicated a clear richness peak in mid-latitude regions (Supplementary Fig. 3f). Mid-latitude regions had higher ppSH richness for both exclusive and non-exclusive phytopathogens (Supplementary Fig. 3g, h). Similarly, the diversity of

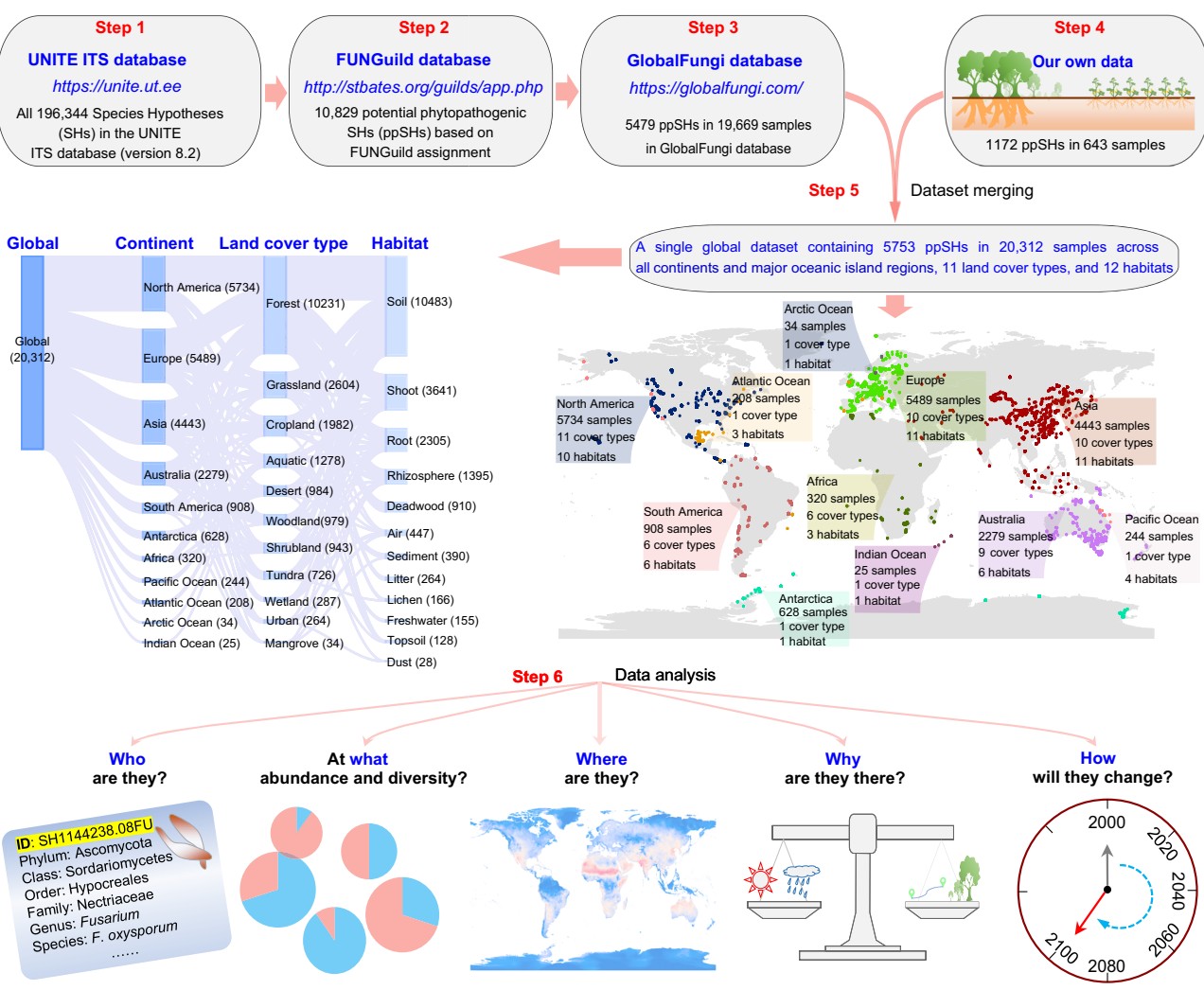

**Fig. 1 | Flow diagram of the methods implemented in the current study.** Step 1: reference sequences and taxonomic information extraction. We extracted all 196,344 reference "species hypotheses" (SHs at 98.5% similarity; SHs can be treated as fungal species) from the UNITE fungal database (version 8.2)[51]. Step 2: construction of reference database of phytopathogenic fungi. The 196,344 SHs were compared against the FUNGuild database (http://www.stbates.org/guilds/app.php; accessed September 2019)[6], and those assigned to 'probable' and 'highly probable' 'plant pathogen(s)' were retained. This yielded a robust reference database containing 10,829 potential phytopathogenic SHs (ppSHs). Step 3: extracting data on the global distributions of phytopathogenic fungi. We used the 10,829 ppSHs to query the GlobalFungi database[25] and extracted the global spatial occurrence data (and associated metadata) for these fungi. This produced a dataset containing 19,669 samples with 5479 ppSHs. Step 4: Processing novel phytopathogenic fungi sequence data. We collected additional data from forest and cropland ecosystems,

sampling plant shoots, plant roots, soils and the rhizosphere during 2017–2021. ITS amplicon sequences were generated and analysed from each of these samples following the methods used for GlobalFungi[25], yielding an additional 643 samples with 1172 ppSHs for inclusion in this study. Step 5: combining existing and novel phytopathogenic fungi distribution data. The published data extracted from GlobalFungi (Step 3) and our new data (Step 4) were merged into a single global dataset containing 20,312 samples with 5753 ppSHs. Step 6: Data analysis. Using the global dataset, we conducted a series of analyses to systematically investigate the global biogeography of phytopathogenic fungi. Briefly, this included examining general biogeography and diversity patterns, universal ecological dynamics, driving forces and predicting future changes of phytopathogenic fungal diversity and distributions in response to climate change. The map was generated in the R language. Source data are provided as a Source Data file.

phytopathogens with exclusively phytopathogenic trophic modes ($R^2 = 0.054$, $F = 571$, Supplementary Fig. 3g) also showed a marginally weaker relationship with latitude compared with than those with non-exclusive trophic modes ($R^2 = 0.066$, $F = 702.6$, Supplementary Fig. 3h). These results confirm that our findings are not biased by the sequencing region, sequencing depth, or trophic mode of phytopathogens. Moreover, to ensure this finding was robust and not biased by differential sampling coverage, we randomly selected 300 (Supplementary Fig. 3i) or 150 (Supplementary Fig. 3j) samples from North America, Europe, Asia, Australia, South America, Antarctica and Africa, which all had >300 samples, and repeated the analysis. This confirmed that the second-order polynomial regression best described the phytopathogen richness peak at intermediate latitudes (absolute latitude ranges

26°–32°) is not driven by sampling effects. Although our resampling approach can, to a great extent, avoid the bias from unbalanced sampling, it's worth noting that resampling from highly dense sampling regions, such as humid areas of East and Southeast Asia in this study, could also potentially produce greater diversity in those areas. Therefore, our results would benefit from further refinement as more data from undersampled regions, especially in the tropics, become available in the future.

## Community structure and spatial turnover of potential phytopathogenic fungi

The ppSH sequences were rarefied to even depth (4000 ppSH reads per sample) to conduct community-level analyses and comparisons. A

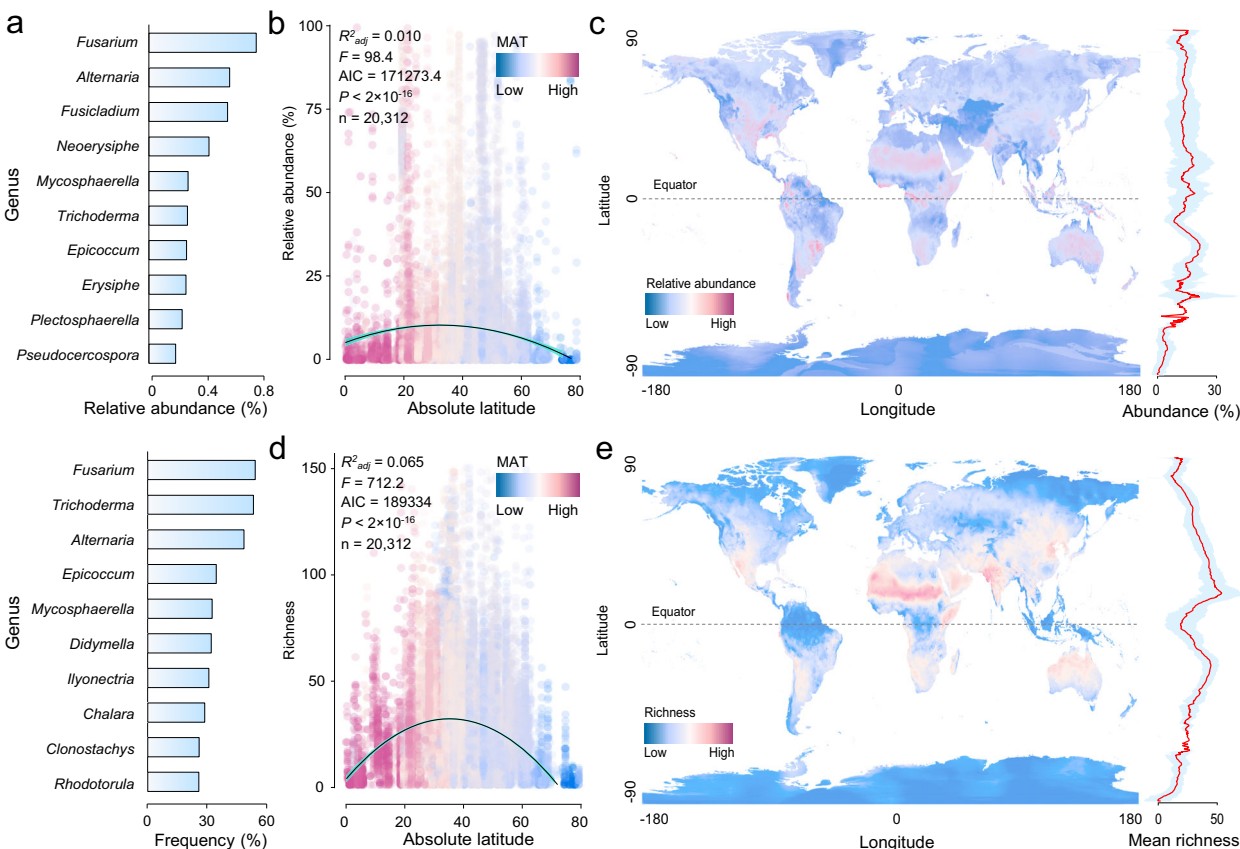

**Fig. 2 | Global biogeography and diversity patterns of potential phytopathogenic fungi. a** Top ten most abundant (top) and frequently occurring (bottom) phytopathogenic fungal genera across 20,312 samples. **b** Latitudinal distribution of relative abundance of phytopathogenic fungi. Colours represent the annual mean air temperatures (MAT) of sampled locations. The line shows the second-order polynomial fit based on ordinary least squares regression. Shaded areas represent the 95% confidence intervals. The analysis was based on one-side *F* and two-side *t* tests (model parameters and *P* values are reported as inset panels). n represents the number of samples. **c** Global relative abundance of phytopathogenic fungi. The relative abundance of phytopathogenic fungi was predicted using GLMs incorporating 19 bioclimatic variables, with the prediction efficiency cross-validated (CV) by common Pearson correlation test using 2/3 samples as a model training dataset and 1/3 as a validation dataset (CV: Pearson *r* = 0.249, *P* < 2 × 10⁻¹⁶, Supplementary Fig. 2a). Climate variables are derived from WorldClim2 at a 5 min

resolution (-10 km). The righthand panel shows the mean (with standard deviation envelope) relative abundance of phytopathogenic fungi across latitudes.
**d** Latitudinal distribution of phytopathogenic fungal species richness. The line shows the second-order polynomial fit based on ordinary least squares regression, and shaded areas represent the 95% confidence intervals. The analysis was based on one-side *F* and two-side *t* tests (model parameters and *P* values are reported as inset panels). n represents the number of samples. **e** Global diversity of phytopathogenic fungi. The global distribution of phytopathogenic fungal species richness was also cross-validated (CV: Pearson *r* = 0.429, *P* < 2 × 10⁻¹⁶, Supplementary Fig. 3). The righthand panel shows the mean (with standard deviation envelope) phytopathogenic fungal species richness across latitudes, with peaks in richness at intermediate latitudes. The maps were generated in the R language. Source data are provided as a Source Data file.

three-way PERMANOVA showed that phytopathogenic fungal communities were compositionally distinct across continents, land cover types and habitats (Supplementary Table 1). Land cover type played the primary role in determining the composition of phytopathogenic fungal communities, followed by sampling region and habitat type (Fig. 3a, Supplementary Fig. 4a and Supplementary Table 1).

A central pattern in macroecology is the distance-decay relationship (DDR), where community similarity decreases as geographic distance between samples increases[34]. DDR reflects spatial community turnover and can also be used to infer underlying ecological processes controlling the community[35,36]. Phytopathogenic fungal communities showed significant DDRs across all land cover types (slope: −0.035 to −0.186, *P* < 0.001) and habitat types (slope: -0.039 to -0.190, *P* < 0.001, Supplementary Fig. 4b). To avoid issues of scale dependency in DDRs, we determined the initial similarity (defined here as the community similarity within one kilometre) and halving distance (distance at which community similarity halves, and thus reflects initial community turnover rates) of phytopathogenic fungal communities from the

observed DDR parameters[36]. Phytopathogenic fungal communities from tundra ecosystems had the highest initial similarity, and those from forests had the lowest similarity (Supplementary Fig. 4c), whereas halving distances were highest in urban and wetland ecosystems but lowest in forests (Fig. 3b). This suggests that forest ecosystems have relatively high spatial turnover of phytopathogenic fungi, most likely reflecting their high heterogeneity in terms of environmental conditions and tree species. Across habitat types, air and plant shoots had the highest and lowest initial similarity, respectively; while topsoil and dust had the highest and lowest halving distance, respectively (Supplementary Fig. 4c, d). This indicates relatively high homogeneity of air-borne phytopathogenic fungi resulting from mixing of propagules from various plant species, habitats and land cover types.

In community variation, two key components exist−turnover (replacement of species) and nestedness (the extent to which species composition of smaller assemblages is a subset of larger assemblages)[37]. In our datasets, compositional changes in the phytopathogenic fungal community across land cover types were

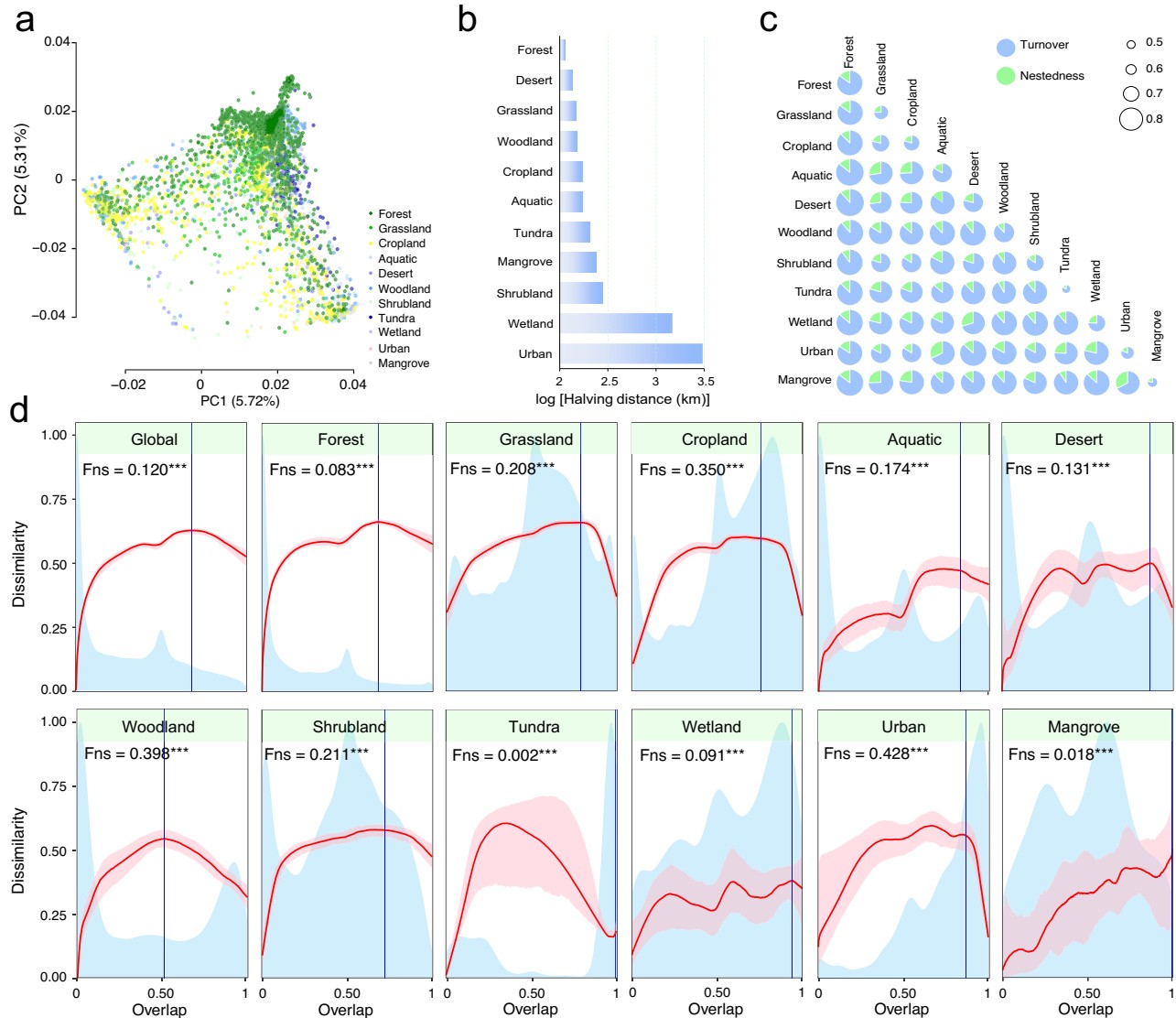

**Fig. 3 | Community turnover and universal dynamics of potential phyto-pathogenic fungi. a** Principal co-ordinates analysis (PCoA) of phytopathogenic fungal communities based on Bray–Curtis dissimilarity. Samples are coloured by land cover type. **b** Halving distance of phytopathogenic fungal community across land cover types. Halving distance is the geographic distance between samples at which community similarity halves (see "Methods" section). **c** Species turnover and nestedness of phytopathogenic fungal communities across land cover types. Circle sizes reflect total beta diversity (measured by Sørensen's distance), with light blue and red colours respectively showing the relative contribution of turnover and nestedness to total community variation. **d** Universal ecological dynamics of phytopathogenic fungi. The ecological universality of phytopathogenic fungi was assessed using dissimilarity-overlap curves. Dissimilarity-overlap curves are in red, the distribution density of sample pair overlap is in light blue, and the point at which the curve first becomes negative is marked by a vertical blue line (chosen by median of 1000 bootstraps). The pink shared area indicates the range of the 94% confidence intervals. The fraction of negative slope (Fns) is the fraction of data points where the dissimilarity-overlap curve has a negative slope (***$P < 0.001$). A higher Fns value indicates that the underlying ecological dynamics of a microbiome are largely host-independent. Source data are provided as a Source Data file.

consistently dominated by species turnover (>66.9%) but not nested-ness (Fig. 3c). For example, species turnover between cropland and other land cover types always exceeded 74% (Fig. 3c), explaining why only a low fraction (40.4%) of ppSHs from croplands can be source-tracked (Supplementary Fig. 5a). Globally, the proportion of shared phytopathogenic fungal sequences across land cover types was con-sistently low (<18%), confirming the relatively low community nest-edness and high turnover (Supplementary Fig. 5b). These findings contrast to global bacterial communities that are characterised by high nestedness[10].

To test for host dependence of phytopathogenic fungi, the Fns (fraction negative slope) index from dissimilarity-overlap curve ana-lysis was employed[38]. In this analysis, a high Fns value indicates that the underlying ecological dynamics of a microbiome are largely host-

independent. In contrast, a low Fns value reflects that the ecological dynamics of a microbiome are host-specific[38]. We observed significant Fns from dissimilarity-overlap curves across the global phytopatho-genic fungal dataset (Fns = 0.12, $P < 0.001$), and independently across all land cover types (Fns range, 0.002 to 0.428, $P < 0.001$; Fig. 3d), and all habitats (Fns range, 0.002-0.211, $P < 0.001$; Supplementary Fig. 6). The Fns of global phytopathogenic fungi observed in the current study (0.12) was lower than those reported for human-associated, bacterial microbiomes (0.23–0.99)[38,39], and lower than those reported for all fungi (0.63)[40] and fungi with other trophic modes such as AM fungi in natural and agricultural fields (0.28–0.94)[41,42]. This suggests that the ecological dynamics of phytopathogenic fungi were potentially more host-specific than other microbial groups. However, given the small number and scope of these studies, the Fns values for fungi need to be

evaluated across more complex ecosystems and at larger scales, as well as via controlled experimental manipulation of factors driving host effects. The relatively high Fns values in cropland (0.35), grassland (0.208), woodland (0.398), shrubland (0.211) and urban (0.428) land cover types, and in soil (0.206) and root (0.211) habitats indicated relatively lower host dependence across these land cover types and/or habitats[38,41].

## Factors determining the global distribution of potential phytopathogenic fungi

Random forest models were constructed to examine the factors determining the global distribution of phytopathogenic fungi. Climatic (indicated by 11 temperature-related and 8 precipitation-related bioclimatic variables), spatial (indicated by longitude, absolute latitude and standardised principal coordinate of neighbour matrices) and vegetation (indicated by gross primary production and plant diversity) variables were separately or jointly considered in six random forest models (Model 1: Climate; Model 2: Space; Model 3: Vegetation; Model 4: Climate & Space; Model 5: Climate & Vegetation; Model 6: Climate & Space & Vegetation; Fig. 4a), and the model performances ($R^2$) were consequently compared.

The global ppSH richness was robustly explained solely by climate factors (Model 1, 71.15%), which displayed better performance than space (Model 2, 65.89%) and vegetation (Model 3, 66.59%) (Fig. 4a). Moreover, adding either space (Model 4, 73.40%) or vegetation (Model 5, 71.80%), or both space and vegetation (Model 6, 73.47%) to a climate-

only model provided only a minor improvement to the explained variation in global ppSH richness (Fig. 4a). Across all variables, 67.19% of the explained variability in global ppSH richness was attributed to climate factors, 16.13% to spatial variables and 15.68% to vegetation-related variables (Fig. 4b). This is consistent with previous studies on both all fungi and on phytopathogenic fungi, whose global distributions were also mainly determined by climatic factors[1,13,17]. For different land cover types, the ppSH richness in forest (68.58%), grassland (78.04%), cropland (75.89%), tundra (60.53%) and urban (61.89%) areas was also best explained by bioclimatic variables. The ppSH richness of phytopathogens with non-exclusive trophic modes was consistently better explained by our random forest models with either solely bioclimatic variables or with the addition of spatial and vegetation variables, both globally and across main land cover types including forest, grassland, cropland, desert, woodland, shrubland, tundra, wetland and mangrove (Fig. 4a). However, climate factors consistently contributed most to the explained variability of ppSH richness across all land cover types and ppSHs with different trophic modes (Fig. 4b). All random forest models showed weaker performance in explaining the relative abundance of ppSHs compared to their richness (Supplementary Fig. 7). Globally, <40% of the variation in relative abundance was explained by the random forest models (Supplementary Fig. 7). While vegetation has been frequently considered to be the key factor driving the distribution of phytopathogens[17], it played a relatively small role globally compared to bioclimatic variables in determining both the diversity and relative abundance of ppSHs, possibly because plant

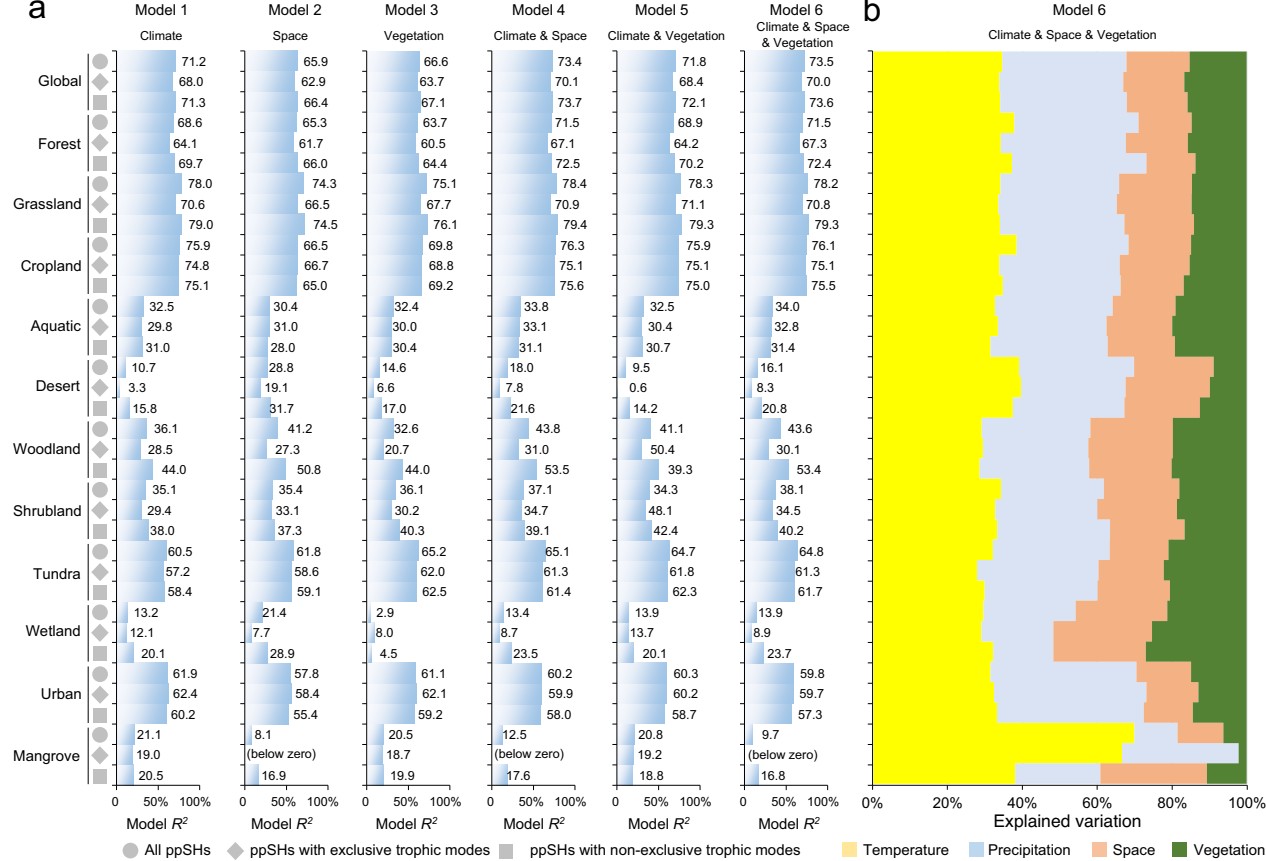

**Fig. 4 | Factors determining the global distribution of potential phytopathogenic fungi. a** Random forest model performances for ppSH richness. **b** Contribution of climatic, spatial and vegetation variables to the explained variation by the complete random forest model for ppSH richness. Each tree was fitted based on a random sample of two-third of the observations ("in-bag"), and each tree

split was based on a different random subset of one-third of the predictors, while the results were cross-validated against the remaining observations ("out-of-bag"), which is in line with standard protocols. The model performance was assessed based on model $R^2$ with 999 permutations. Vegetation reflects gross primary production and plant diversity. Source data are provided as a Source Data file.

distributions are also highly determined by temperature and precipitation at the global scale[43]. However, climate change might indirectly affect the distribution of phytopathogens through modifying the composition of host-plant communities, which may provide an additional dimension to the effects of future climate on the distribution of phytopathogens. Moreover, we acknowledge that our vegetation index only considers plant biomass (gross primary production) and overall plant diversity rather than plant species identity and taxonomy, which were largely unavailable for this study. Nonetheless, these key factors may determine the distribution of phytopathogens, especially at local and regional scales[44,45], once the overarching influence of temperature and precipitation on shaping the global distribution of biomes has been accounted for.

## Global diversity and invasion risk of potential phytopathogenic fungi under future climate change scenarios

The effects of climate change on the diversity and distribution of phytopathogenic fungi (across land cover types and habitats) has remained a major uncertainty. Here, we modelled phytopathogenic fungal diversity and relative abundance in the year 2100 under four future climate scenarios (Shared socioeconomic pathway (SSP); sustainability (SSP126); middle of the road (SSP245); regional rivalry (SSP370); and fossil-fuelled development (SSP585) scenarios) using eleven different CMIP6 downscaled global change models (GCMs). Our projections showed consistent increases in phytopathogenic fungal diversity under all future climate scenarios compared to current climate conditions across all land cover types and habitats (13.32−30.43%, Fig. 5a). The regions with predicted increasing diversity account for more than 90% of the global area, especially in the Arctic where phytopathogenic fungal diversity is expected to increase sharply (Fig. 5a). These results were consistent when data were analysed for phytopathogenic fungi with both exclusively phytopathogenic trophic modes and those with non-exclusive trophic modes (Supplementary Fig. 8a). However, the richness of phytopathogens with exclusive trophic modes is expected to have a greater increase than phytopathogens with non-exclusive trophic modes under all future climate scenarios and under all eleven GCMs (Supplementary Fig. 8a). The ppSH richness increase in grassland was relatively slight (2.14−3.32%), but we observed the largest increases in phytopathogenic fungal diversity in forest (37.27−79.12%) and cropland (34.93−82.51%) ecosystems and in soil (11.95−28.19%), plant shoot (38.63−96.14%) and root (29.99−78.16%) habitats across climate change scenarios (Fig. 5a). We also predicted that the relative abundance of ppSHs in forest ecosystem and soil habitat will increase (all samples: 0.5−1%; forests: 1.49−3.92%; soils: 0.43−1.03%; Supplementary Fig. 9). However, not all phytopathogenic fungi showed a consistent increase at the genus level (Supplementary Fig. 9). For the top ten most relatively abundant genera, seven (*Fusarium, Alternaria, Acrodontium, Trichoderma, Epicoccum, Erysiphe* and *Pseudocercospora*) were expected to increase in relative abundance, while the other three genera (*Fusicladium, Neoerysiphe* and *Plectosphaerella*) were predicted to show opposite trends (Supplementary Fig. 9). It is important to note that these predictions are based on observational data alone, with no mechanistic inference to drive these trends. Moreover, our projections are founded on a permanent climate-diversity/relative abundance relationship, and the projections may need to be amended if climate-diversity/relative abundance relationships change under future climate scenarios. For example, if in the future, extremely high temperatures experienced in some regions exceeds the thermal limits for growth of many phytopathogenic fungi, our prediction will fail for those regions. Our predictions may also be biased if major land use or vegetation type changes occur. Nevertheless, our predictions are consistent with previous research, which predicted an increased proportion of soil-borne pathogens (~0.8% to 2.3%) under warming conditions in global soils[17], and thus lends additional inference to support this global trend.

These predicted major changes in phytopathogenic fungi have the potential to threaten global carbon storage, food security and ecosystem sustainability. Recent research suggests that climate change will lead to 243% (SSP126) to 460% (SSP585) increase in the occurrence of crop pests and diseases (CPD) by the end of this century[24]. The close match between ppSH richness and CPD emergence in croplands implies that for each 1% increase in phytopathogenic fungal diversity, CPD emergence would potentially increase by 2.25% (Pearson's $r = 0.405$, $P = 0.002$, Supplementary Fig. 8b). It should be noted that direct links between the diversity and relative abundances of ppSH and the incidence of plant disease are not yet established. For example, high phytopathogen diversity could imply greater disease incidence with a broader range of potential hosts affected, but may also lead to lower disease risk due to dilution effects and lower densities of host-specific fungal propagules. Furthermore, pathogen diversity may be a key regulator maintaining the plant diversity by relatively stronger suppression of dominant species, hence preventing competitive exclusion[46,47]. Therefore, the observed positive correlation between ppSH diversity and CPD emergence in croplands may not hold in other land cover types, and caution is required when extrapolating the ppSH-disease relationships of anthropogenic habitats to natural ecosystems. Moreover, diversity and relative abundances may be unrelated to absolute abundances, which could not be deduced from our datasets. For example, high phytopathogen diversity with low absolute abundance may not bring greater disease risk compared with low phytopathogen diversity with high absolute abundances. Therefore, further experimentation is required to fully examine the mechanisms underpinning the diversity-disease severity relationship, and the contrasting responses of phytopathogenic fungal diversity from different land cover types and habitats to climate change.

Using a maximum entropy model[48], we assessed the current global invasion risk of all phytopathogenic fungi taken together (Fig. 5b and Supplementary Fig. 10), and predicted their invasion risk under future climate-change scenarios using the eleven CMIP6 GCMs (Supplementary Fig. 10a). Analysis of the receiver operating characteristic curve revealed that all predictions show high accuracy based on the high AUC values (AUCs > 0.890, Supplementary Table 2). Under current and all reasonable future climate scenarios, eastern Asia, Europe, southern Africa, southern Australia, southern South America and central North America have a relatively higher invasion risk of phytopathogenic fungi compared with other regions (Supplementary Fig. 10a). On average, all eleven GCMs showed an increased risk of invasion of phytopathogenic fungi under all future climate scenarios (3.3−5.4%, Fig. 5b). The relative change in invasion risk was significantly positively correlated with the predicted change in ppSH richness ($r = 0.434$, $P = 0.004$, Supplementary Fig. 8c).

Globally, the areas with increased risks of invasion from phytopathogenic fungi are mainly distributed at mid-latitudes (Supplementary Fig. 10b), which is associated with the higher ppSH richness of these regions. Although the mean invasion risk value increases under SSP126 (3.3%), relatively less area (45.9%) shows an increased risk of invasion from phytopathogenic fungi under sustainable development projections (SSP126; Fig. 5b). While under the other three scenarios, a greater area shows an increased risk of invasion in the future, especially under projections of fossil-fuel dominated scenarios (SSP245: 61.4%; SSP370: 54.6%; SSP585: 67.2%; Fig. 5b). It's worth noting that the polar regions, which are likely to show the greatest increases in temperature under climate change, may also suffer from an increased risk of invasion from phytopathogenic fungi under SSP585 (Supplementary Fig. 10b). This is consistent with the currently observed range shifts towards higher latitudes in many pathogenic species[49,50]. Therefore, our results highlight the need to further reduce global greenhouse gas emission in order to limit future phytopathogen invasion, which could be especially prominent in mid- and high-latitude regions.

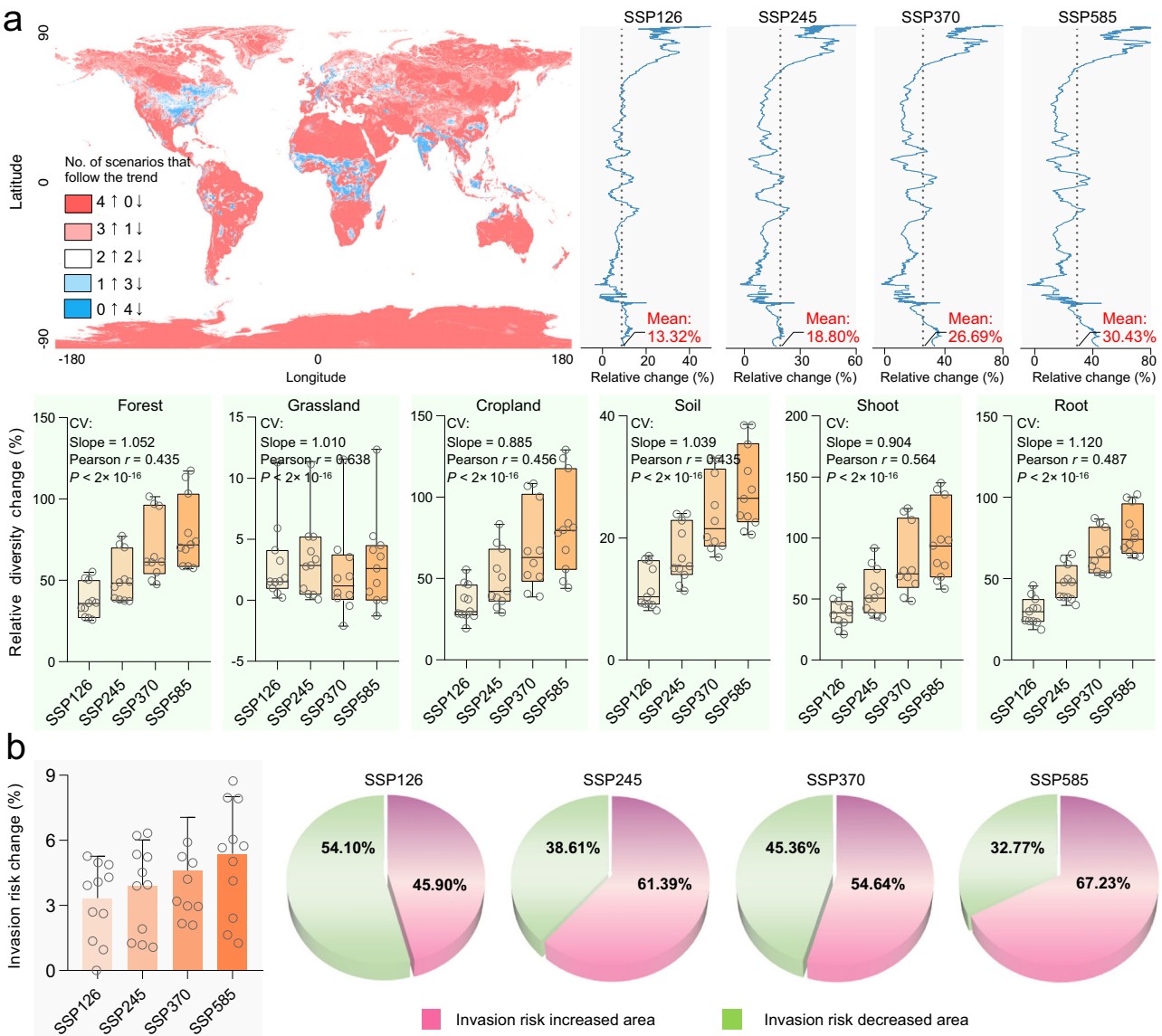

**Fig. 5 | Diversity and invasion risk of potential phytopathogenic fungi by 2100 under future climate change scenarios. a** Predicted change in phytopathogenic fungal diversity under future climate-change scenarios. A diversity-climate model was constructed by GLMs using ppSH richness and 19 climate variables. This model was used to predict future ppSH richness across all land cover types, or within main land cover types, or from main habitats, under four different climate scenarios. Predictive models were cross-validated (CV) by common Pearson correlation test using 2/3 samples as a model training dataset and 1/3 as a validation dataset. All climate variables were derived from WorldClim2 using a 5 min (~10 km) resolution. The future climate data were derived from eleven different CMIP6 downscaled global change models (GCMs; See detailed information in Methods. NB−The climate data of model FIO-ESM-2-0 under the SSP370 scenario in 2080-2100 are not available). The relative change in ppSH richness under different GCMs compared to current climate conditions were averaged. Plot axis labels reflect, shared socio-economic pathway (SSP); sustainability (SSP126); middle of the road (SSP245); regional rivalry (SSP370); and fossil-fuelled development (SSP585) scenarios. Box plots indicate median (middle line) with 25th, and 75th percentile (box), and 5th and 95th percentile (whiskers). $n = 11$ for SSP126, SSP245 and SSP585; $n = 10$ for SSP370. n represents the number of GCMs. The map was generated in the R language. **b** Mean invasion risk under different future climate scenarios. After constructing the invasion-climate model, the future invasion risks were predicted using the climate data derived from eleven different CMIP6 downscaled GCMs, and the invasion risks under different GCMs compared to current climate were averaged (mean ± s.d.; $n = 11$ for SSP126, SSP245 and SSP585; $n = 10$ for SSP370. n represents the number of GCMs). The pie charts represent the increased/decreased invasion risk area. Source data are provided as a Source Data file.

## Limitations, conclusions and future perspectives

This study provided an omnidirectional understanding on the global diversity and biogeography of phytopathogenic fungi. It is important to acknowledge that using closed-reference data, and lacking a reference database fully resolved to species level, may obscure some of the overall diversity and abundance of phytopathogenic fungi. However, the distributions and biogeographic patterns of phytopathogenic fungi presented here are still valuable, and provide a robust first-order approximation of their underlying ecology and susceptibility to climate change. In addition, $CO_2$ fertilisation effects were not considered in this study's predictions of future changes in phytopathogenic fungi, as the direct relationship between $CO_2$ concentration and phytopathogens remains ill-defined. Furthermore, without global data on the range distribution of plants, directly linking fungal pathogens with actual plant host distributions remains challenging. Our observational data can only provide correlative insights into the distributions of plant pathogens, but a mechanistic understanding of current and future trajectories of pathogen biogeography will require detailed information about host plant ranges. Further research into the distribution of plant species and their niche ranges will be fundamental to

facilitating the joint species distribution modelling that is needed to assess these trends in the future.

Our findings highlight the importance of climate factors in determining the diversity and composition of phytopathogenic fungal communities across the globe. More importantly, our model projections indicate a potential increase in phytopathogenic fungal diversity, and an increased global invasion risk especially under unsustainable development scenarios. Such increased fungal pathogen risks are likely to have direct consequences for the productivity and sustainability of managed and natural ecosystems, with direct implications for food production and carbon sequestration. Our research suggests that low-emission-dependent climate scenario can mitigate the invasion risks caused by climate changes, shedding light to optimised sustainable development solutions that protects agriculture and forestry from the harmful impacts of phytopathogens.

## Methods

### Data collection and processing

**Constructing a reference database of phytopathogenic fungi.** We extracted all 196,344 reference species hypotheses (SHs) from the UNITE fungal database (version 8.2)[51]. The 196,344 SHs were queried against the FUNGuild traits database (http://www.stbates.org/guilds/app.php; accessed September 2019)[6], and those assigned to the guild 'plant pathogen' with confidence of 'probable' and 'highly probable' were retained. This yielded a reference database containing 10,829 potential phytopathogenic SHs (ppSHs).

**Processing new data.** New samples were collected across East Asia from forest and cropland ecosystems, with samples from plant shoot, plant root, soil and rhizosphere habitats during 2017–2021. After extracting total DNA, the ITS region was amplified using the fungal PCR primers ITS1F (5′-CTTGGTCATTTAGAGGAAGTAA-3′) and ITS2 (5′-GCTGCGTTCTTCATCGATGC-3′)[52] and then sequenced on an Illumina MiSeq platform. Sequencing data were analysed following the methods described in GlobalFungi[25]. Briefly, the ITS sequences were extracted using ITSx (v1.0.11)[53], and non-ITS sequences were removed. The extracted ITS sequences were classified according to the representative sequence of the closest UNITE species hypothesis (SH) using BLASTn[54] with a 98.5% similarity threshold. Representative sequences of each SH were queried against the reference database constructed above to identify ppSHs. This yielded a dataset containing 643 samples (from forest and cropland ecosystems, and plant shoot, root, soil and rhizosphere habitats) with 1172 ppSHs.

**Meta-analysis data collection and trimming.** The 10,829 ppSHs were queried against the GlobalFungi database (3rd release, January 5, 2021; https://globalfungi.com/)[25], and the targeted SHs with associated metadata including sample source, geographical location, land cover type and habitat type, sampling time, abundance and total sequencing depth were downloaded using the *Taxon search* function. After merging the data by sample IDs, we produced a dataset containing 19,669 samples with 5479 ppSHs.

Finally, the published data in GlobalFungi and our new sequenced data were merged into a global dataset containing 5753 ppSHs from 20,312 samples across all seven continents and four major oceanic island regions, 11 land cover types (forests, grasslands, croplands, aquatic, deserts, woodlands, shrublands, tundra, wetlands, urban and mangroves) and 12 habitat types (soils, plant shoots, roots, rhizosphere, deadwood, air, sediment, litter, lichen, freshwater, topsoil and dust (atmospheric deposition)) (Fig. 1). The number of ppSH sequences was divided by the total ITS sequences of each sample to calculate the relative abundance. The total relative abundance of the potential plant pathogens was highly correlated with the same variable calculated using a rarefied OTU table (8000 reads per sample; $r = 0.999$, $P << 0.001$), so the choice of not rarefying our data did not affect our

conclusions. We also tested the deviation between close-reference-based and open-reference-based outputs using our new data. The results showed a perfect match between the relative phytopathogen abundance of close-reference-based and open-reference-based outputs ($r = 0.983$, $P << 0.001$), suggesting that using close-reference-based data would not quantitatively affect our conclusions. All *Gibberella* reads were considered as *Fusarium* in this study for consistency with the most recent classifications[55].

### Climate factors and vegetation data

Nineteen bioclimatic variables for each sample's location were extracted from WorldClim2 (https://www.worldclim.org/)[56]. The historical climate data represent the average for the years 1970-2000 and comprise 19 variables, 11 of which are temperature-related, and 8 of which are precipitation-related (for detailed information see Supplementary Table 3; https://www.worldclim.org/data/bioclim.html). The future climate data (2080-2100) are CMIP6 (Coupled Model Intercomparison Project 6, https://esgf-node.llnl.gov/projects/cmip6/) downscaled future climate projections. Monthly values of minimum temperature, maximum temperature and precipitation were processed for four Shared Socio-economic Pathways (SSP): 126, 245, 370 and 585 (SSP126: sustainability; SSP245: middle of the road; SSP370: regional rivalry; SSP585: fossil-fuelled development)[57]. The climate data under different SSP scenarios were separately predicted using eleven CMIP6 downscaled global change models (GCMs), namely ACCESS-ESM1-5, CanESM5, CanESM5-CanOE, CNRM-CM6-1-HR, CNRM-ESM2-1, EC-Earth3-Veg, FIO-ESM-2-0, GISS-E2-1-G, MIROC6, MRI-ESM2-0 and UKESM1-0-LL. The vegetation variables capture gross primary production (GPP) and overall vascular plant diversity. The GPP data used in this study were the annual average GPP data during the last four decades derived from satellite near-infrared reflectance data[58]. The plant diversity data were extracted from the global map of alpha diversity (local species richness, 1 km resolution) for vascular plants built from 170,272 georeferenced local plant assemblages[59].

### Statistical analysis

**Global alpha-diversity fitting.** Richness (defined as the number of observed ppSHs in this study) was used to measure taxonomic α-diversity of phytopathogenic fungi. Two approaches were used to investigate the global diversity patterns of phytopathogenic fungi: generalised linear models (GLMs) and ordinary least squares regression[13]. First, we used the generalised linear models (GLMs) to fit global ppSH diversity data. Briefly, we used GLMs to select the variables that were significantly related to diversity from bioclimatic variables BIO1-BIO19, because these variables are broadly considered to influence fungal diversity, and have been used to predict diversity patterns of multiple microbial groups at various scales[1,60,61]. We predicted the global diversity pattern of phytopathogenic fungi after characterising the environment within each grid cell under the resolution of 5 min (~10 km). A cross validation was conducted to test the efficiency of our prediction. Briefly, 2/3 of samples were randomly selected as the modelling dataset, while the remaining 1/3 samples were selected as the validation dataset. After constructing the GLMs using the modelling dataset, we predicted the diversity of phytopathogenic fungi in the validation dataset. Predicted diversity was plotted against the observed diversity, and the correlation coefficients used to determine the prediction efficiency[62]. Second, the diversity-latitude relationship was fitted using simple linear regression and second-order polynomial regression to infer if a latitudinal diversity gradient (LDG) exists. We also tested if the richness-latitude relationship was not biased by the different sequencing region (ITS1, ITS and both) by separating the samples according to sequencing region information (Supplementary Fig. 3e). To confirm the richness-latitude relationship was robust and not biased by unequal sequencing depth, we equally and randomly chose 4000 ppSH reads for each sample, and

calculated the richness-latitude relationship (Supplementary Fig. 3f). Given the potential unbalanced sampling effect, we randomly selected 300 or 150 samples in North America, Europe, Asia, Australia, South America, Antarctica and Africa where >300 samples were collected. The diversity-latitude relationship based on subsampled data was fitted using both simple linear regression and second-order polynomial regression, and the fitting parameters including model $R^2$, $F$ values, AIC and $P$ values were compared. Both random resampling procedures (300/150 samples) were conducted 100 times in R language, and the comparisons were also conducted 100 times to avoid the randomisation bias (Supplementary Fig. 3i, j).

**Community-level analyses.** The ppSH sequences were rarefied to even depth (4000 sequences per sample) to conduct community-level comparisons and analyses. Bray–Curtis (abundance-based) and Sørensen (occurrence-based) distances were calculated to quantify taxonomic β-diversity using the *vegan* R package. Three-way PERMANOVA was conducted to investigate the main determinants of variation in species composition. The variation in species composition can be further divided into two components, species turnover (species replacement by others) and nestedness (the extent to which the species composition of small assemblages is a subset of larger assemblages)[37,63]. The *betapart* R package was used to disentangle the relative contribution of species turnover and nestedness to the β-diversity[37].

**DDRs and halving distance.** To determine the spatial turnover of phytopathogenic fungi, the distance-decay relationship (DDR) was calculated as the slope of a linear least squares regression on the relationship between log-transformed geographical distance [log(distance+1)] and community similarity (Bray–Curtis similarity). The DDR slope was calculated following the formula below.

$$\log S = a + b\log(D+1) \tag{1}$$

where $a$ is the intercept and $b$ is the slope of the distance-decay relationship. $S$ and $D$ are community similarity and geographical distance, respectively. The significance of the DDR slope was determined using matrix permutation test (999 permutations). Given the scale-dependency of DDRs, we calculated the initial similarity (similarity within one km) and halving distance (HD, the distance at which community similarity halves) to eliminate scale-dependent errors[36]. Briefly, the initial similarity was calculated following the formula below.

$$S(1) = a + b \times \log 2 \tag{2}$$

While the HD was calculated as follow.

$$HD = 10^{\frac{(b \times \log 2) - a}{2b}} \tag{3}$$

**Shared sequences and source tracking analysis.** The proportion of shared sequences reflects species turnover across different systems[64]. Therefore, we calculated the pairwise shared proportion of sequence numbers, and used the average shared proportion as a proxy for species turnover. To determine where the phytopathogenic fungi in different land cover types come from, we estimated the source of phytopathogenic fungi in different land cover types using SourceTracker[65], which was run in the R language with default settings at genus level.

**Universal ecological dynamics.** A dissimilarity overlap curve analysis was conducted to assess whether the ecological dynamics across phytopathogenic fungal communities are universal[38,40] using R package DOC. The dissimilarity-overlap curve emerges by plotting the community dissimilarity against the fraction of taxa that overlap. Where the curve dips as the overlap grows, universality is supported and the level of support is proportional to the fraction of pairwise comparisons under the graph where the curve slope is negative (termed the fraction negative slope, Fns). A high Fns value indicates that the underlying ecological dynamics of a microbiome are largely host-independent. In contrast, a low Fns value reflects that the ecological dynamics of a microbiome are host-specific[38]. For the smoothed curve of a given dissimilarity-overlap curve, the initiation of negative slope represents the median of initiation of negative slopes calculated from dissimilarity-overlap curves of 1000 bootstrapped data sets.

**Random forest modelling.** We applied a machine-learning random forest model to quantitatively examine the key variables influencing the relative abundance and diversity of phytopathogenic fungi using the *randomForest* R package[66]. Six random forest models were constructed. Climatic (indicated by 11 temperature-related and 8 precipitation-related bioclimatic variables), spatial (indicated by longitude, absolute latitude and standardised principal coordinate of neighbour matrices) and vegetation (indicated by gross primary production and plant diversity) variables were separately or jointly considered in six random forest models (Model 1: Climate; Model 2: Space; Model 3: Vegetation; Model 4: Climate & Space; Model 5: Climate & Vegetation; Model 6: Climate & Space & Vegetation). To reduce collinearity among predictors, we reduced the initial set of 24 predicting variables to 15 variables with a variation inflation factor (VIF) below 10. This final set included ten bioclimatic variables (BIO2: Mean diurnal range; BIO3: Isothermality; BIO4: Temperature seasonality; BIO8: Mean temperature of wettest quarter; BIO9: Mean temperature of driest quarter; BIO13: Precipitation of wettest month; BIO14: Precipitation of driest month; BIO15: Precipitation seasonality; BIO18: Precipitation of warmest quarter; BIO19: Precipitation of coldest quarter), three spatial variables (longitude, absolute latitude and standardised principal coordinates of neighbour matrices (PCNM)) and two vegetation variables (gross primary production and plant diversity). The variation inflation factor values of predictors are listed in Supplementary Table 4. Five hundred trees were fitted in each model. Each tree was fitted based on a random sample of two-thirds of the observations ("in-bag"), and each tree split was based on a different random subset of one-third of the predictors, while the results were cross-validated against the remaining observations ("out-of-bag"), which is in line with standard protocols[66]. The model performance was assessed based on model $R^2$ using *rfUtilities* R package with 999 permutations. To express variable importance across all modelled ppSHs, the relative importance of each predictor was calculated as a sum of predictor relative importance of all Random Forests for ppSH richness/relative abundance weighted by Random Forest predictive ability (out-of-bag $R^2$)[13].

**Future diversity, relative abundance and invasion risk projections.** As described above, the global diversity pattern under current climate was estimated using GLMs and then cross-validated. Using the model constructed based on the current climate data, the global diversity pattern under different future climate scenarios were then estimated based on the model parameters. For detailed land cover types and habitats, the future diversity patterns were only predicted for those with >1500 globally distributed samples to avoid issues of low sample coverage. We obtained different diversity projection equations for each of land cover type and habitat, and all projections were cross-validated following the procedures described above. We predicted the future diversity under different climate scenarios using the climate data derived from the above eleven different CMIP6 downscaled GCMs, and the relative diversity changes were averaged. The procedures predicting the future relative abundance were same. The projections were conducted using the formula listed in Supplementary Table 5.

We then used the maximum entropy model[48] to make a global invasion risk projection of phytopathogenic fungi under different future scenarios using the Maxent software, which has been widely used to forecast species distribution probability of various macro and microbial organisms at the global scale[67–70]. The standardised outputs indicate predicted probability that conditions are suitable, with higher values indicating high probability of suitable conditions for the species, and lower values indicating low predicted probability of suitable conditions. Here, we used the predicted probability as an agent of invasion risk. First, the occurrence data of phytopathogenic fungi and current climate data were used to predict the global distribution probability of phytopathogenic fungi with 30% of the samples as the random seed. Second, based on the constructed invasion-climate model, the future climate data derived from eleven CMIP6 downscaled GCMs were used to predict the future invasion risk. Third, the relative change in invasion risk under future climate compared to current climate was calculated in AcrMap (version 10.2, https://www.esri.com/)[71] using the Raster Calculator tool. The receiver operating characteristic curve (ROC) analysis was simultaneously conducted to assess the prediction efficiency.

## Reporting summary

Further information on research design is available in the Nature Portfolio Reporting Summary linked to this article.

## Data availability

All raw data used in the current study including reference database, sample metadata, climate data and species-abundance dataset are publicly available in Figshare[72]. The newly generated sequence data are available in the NCBI Sequence Read Archive (SRA) under the accession number PRJNA1021497. The UNITE fungal database is available in https://unite.ut.ee/. The samples and the corresponding metadata in GlobalFungi database are available in https://globalfungi.com/. The current and future climate data are available in WorldClim2 (https://www.worldclim.org/). Source data are provided with this paper.

## Code availability

Most numerical analyses included in this article do not have an associated code. Used codes are available in Figshare[72].

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

## Acknowledgements

We thank the authors whose data were used in this global synthesis. This study was supported by the Fundamental Research Funds for the Central Universities (KYZZ2023003 to J.J.; KYQN2023027 to P.L.; XUE-KEN2022003 to B.W.), National Natural Science Foundation of China (42207349 to P.L.; 41977056 to B.W. and 42107336 to L.L.), Natural Science Foundation of Jiangsu Province (BK20221005 to P.L.), Jiangsu Funding Program for Excellent Postdoctoral Talent (2022ZB331 to P.L.), China Postdoctoral Science Foundation (2022M711653 to P.L.) and grants from D.O.B. Ecology and the Bernina foundation to T.W.C.

## Author contributions

J.J., A.J.D. and B.W. designed the framework. P.L., M.W., M.L., J.L., D.L. and Y.L. contributed to the sample collection. P.L., L.T., L.L., T.L., K.L. and S.W. performed the data analysis. P.L., L.T., B.W., S.Y., T.W.C., M.S., L.L., A.J.D., Z.L. and J.J. wrote the paper. All authors discussed the results and commented on the manuscript.

## Competing interests

The authors declare no competing interests.

## Additional information

Baozhan Wang, Alex J. Dumbrell or Jiandong Jiang.

