## [Peer Review File · Nature Communications]

Global diversity and biogeography of potential phytopathogenic fungi in a changing worldREVIEWER COMMENTS

Reviewer #2 (Remarks to the Author):

I appreciate that the authors have tried to streamline the story, and the manuscript is definitely more readable than the last version. However, my two main concerns (that the other reviewers were clearer in articulating, so I now adopt some of their wording) were (1) that the potential phytopathogens might not all be phytopathogenic and (2) that plant host range was not considered. Unfortunately I do not think that these concerns were alleviated by the authors new analyses. Of course, I understand that these are very difficult to address, but I was disappointed that authors still do not consider how these issues might influence their results.

For the first issue, the authors now compare patterns of potential phytopathogens to those that consider exclusively pathogenic. They do not elaborate on this comparison (Supp Fig 2C & D) or even test if they are the same distribution. However, my understanding is that because they are arguing that there is little difference between the patterns of two categories. However, this result makes me wonder whether this distribution is the same for all fungi? Or does it suggest that even the exclusive phytopathogens are not really phytopathogens? Is the study is mainly capturing likely changes in fungal abundance generally? There is no discussion about this.

For the second issue, while I appreciate that data on particular plant hosts would be another project in and of itself, plant biomass is a poor indicator of plant host range. This caveat needs to be discussed. Despite several of the reviewers bringing this up, it does not seem that the new manuscript even includes the word "host" anywhere. So this caveat is not addressed.

Finally, more generally, there are still lots of analyses that are not set in a context to understand what it all means. In particular, it would be helpful if more of the discussion could point out how the patterns differ from all fungi. How are potential phytopathogens different from all fungi? Or are they the same? (A specific example of a result with no context is the newly added analyses about dissimilarity-overlap curve (Line 203). This is not

a well-known metric in the field and would need to be described better. After looking at the citation, I was not surprised that this is the case for environmental habitats. If this metric is used, I would suggest the authors specify their hypothesis and alternatives that might make sense. As a stand alone, it is hard to interpret what this means for the future of phytopathogens, and how this might compare to any other type of fungi.)

In sum, while the manuscript presents some interesting analyses, it still is not accompanied with a clear interpretation about what the patterns mean biologically, let alone what the limitations of the study are.

Minor comments:

Dust habitat. In the rebuttal letter, it is mentioned that this is atmospheric deposition, but this should be defined in the text, too, as it is used throughout.

Line 191: "two predominant components" – this is strange wording. Nestedness and turnover are certainly two metrics of community variation, but this suggests somehow that they are dominant (or that we rank different metrics of beta-diversity).

Reviewer #3 (Remarks to the Author):

The revised version of the manuscript addresses most of the concerns I raised in the previous version, but a few issues remain. There is no doubt that the paper provides a sound comprehensive look at patterns of fungal diversity and relative abundance and concludes that temperature is a major driver of these patterns and that for richness there is a mid-latitude peak in diversity. These are the the most sound conclusions. The authors then use random forest models to look at the factors driving individual taxa and MaxEnt style niche models to predict how distributions might change as a function of temperature as a measure of "invasion" potential. Again, think there is little quibble with the data and major patterns, but there remain some issues of interpretation or extrapolation that are of some concern. These have to do with three of the main points from my initial review that the authors address in their response. My comment not their responses are here:

Point 1. I appreciate the addition of primary productivity (PP) into the models, but this does not really adequately capture the effect of host plant richness and composition, just total production or biomass. I think it is important to acknowledge that there is no real estimate in the model of, for example, how the diversity of plant resources scales with diversity of fungi. PP can be high with either high or low host diversity, so using PP alone doesn't seem to resolve the issue. While it might be true that the range of a particular genus or family is broad, again, I think this is more about the phyletic or functional diversity of hosts within a region. On Line 245, the statement that "vegetation" is of limited importance is grossly overstated. If the authors want to conclude that PP has limited effect that is fine, but that is not the same as saying vegetation doesn't matter. Species composition and local beta diversity of host taxa almost certainly play a role here. The effects of climate might, indeed, act indirectly through plant composition in addition to directly based on abiotic controls. This needs to be acknowledged and also acknowledge that it may complicate effects of future climate.

Point 2. I agree that diversity and relative abundance are useful, but it is important to recognize that the ecological effect of each of these is very uncertain. There is considerable literature that shows diversity affects ecosystem processes, but inferring the direction of those effects with respect to a particular process like disease is a considerable leap in logic that needs better support or more explanation about the potential range of outcomes of a change in diversity. For example high diversity could imply more disease with a broader range of potential hosts affected or lower disease risk due to dilution and lower density of fungal propagules of any one ppSH.

Point 3. I appreciate the resampling approach, however the issue with the approach described is that resampling from all regions that have > 300 samples can be confounded by the spatial distribution of those samples... if some regions have far more than 300 samples then they likely sample a broader range of habitat types and so the resampling approach will (by chance) tend to produce greater diversity in those areas. If there is no relationship between sampling intensity and latitude, then just say that and you can move on. If there is such a relationship, then I think this needs to be mentioned as an important caveat along with the resampling approach mentioned above. I am not saying it negates the conclusions,

just that it needs to be acknowledged. Specifically, I would like to see mentioned the fact that the sampling appears to be biased at the 26-32 degree latitude to humid areas of E and SE Asia and that the unequal longitudinal distribution of latitudinal samples might confound interpretation. Because the authors included longitude a variable in these analyses, it would seem that these effects do not negate the latitudinal patterns (though incorporating longitude is often tricky because of the circular nature of the globe... the authors say they used absolute longitude... does that mean that they used absolute value (0 to 180) or absolute (+180 to -180). If the latter than this seems the best one can do, but if the former, then this needs to be redone since it is hard to imagine why +90 and -90 would be expected to have the same fungal diversity values.

Two additional concerns:

I still do not understand what universal dynamics means. Perhaps this is something known in the fungal ecology world, but as I have never heard this term and it needs a brief explanation in the text (not just in supplemental) if it is to be used here.

I appreciate that current analyses do not allow quantification of absolute abundances of ppSHs and that the present analyses still have value with richness, diversity and relative abundances. That said, I think the authors should acknowledge how knowing patterns of absolute abundance might alter the conclusions of the present manuscript. Total fungal abundance could be assessed and presumably this would scale largely with humidity, at least to a first order approximation. If that seems reasonable, how would knowing that alter the conclusions of the authors regarding the effects of changing climate on fungal effects on plants.

Point-by-point response to reviewers' comments

(NCOMMS-23-00544A)

We appreciate the constructive comments provided by the reviewers and their time spent in carefully reviewing the manuscript. All points have been fully addressed, and we believe this has substantially contributed to improving the overall quality, readability, and logic of the article. We hope that our revisions successfully address any concerns the reviewers raised. We have highlighted our changes in blue in the '*Revised manuscript with marked changes*' file.

Reviewer #2 (Remarks to the Author):

Q1: I appreciate that the authors have tried to streamline the story, and the manuscript is definitely more readable than the last version. However, my two main concerns (that the other reviewers were clearer in articulating, so I now adopt some of their wording) were (1) that the potential phytopathogens might not all be phytopathogenic and (2) that plant host range was not considered. Unfortunately, I do not think that these concerns were alleviated by the authors new analyses. Of course, I understand that these are very difficult to address, but I was disappointed that authors still do not consider how these issues might influence their results.

Response: Thank you very much for your supportive words on the version you reviewed, and we apologise that our initial revisions didn't fully alleviate your concerns. We have now made further modifications to fully address these issues. Briefly, (1) we compared and discussed the differences between phytopathogens with exclusively phytopathogenic modes and non-exclusive modes as well as the influence all fungi may have on observed distribution patterns, driving factors and future changes (Please see our response to your Q2 for details); (2) we provided an exploration of how host-plant ranges may influence phytopathogenic fungal distributions by including global plant diversity data in our analysis. We acknowledge that strictly speaking, plant diversity is different from individual host-plant ranges, but we felt this addition provided the closest approximation of the influence of host-plants on phytopathogenic fungal distributions that currently available data could provide. Nevertheless, we appreciate the reviewer's comment that including this information about plant species

ranges remains a major challenge, and we agree that it is important to discuss and acknowledge the importance of considering REAL host range in predicting the distribution of phytopathogens (Please see our response to your Q3 for details). We include further details below and we hope these modifications are satisfactory.

Q2: For the first issue, the authors now compare patterns of potential phytopathogens to those that consider exclusively pathogenic. They do not elaborate on this comparison (Supp Fig 2C & D) or even test if they are the same distribution. However, my understanding is that because they are arguing that there is little difference between the patterns of two categories. However, this result makes me wonder whether this distribution is the same for all fungi? Or does it suggest that even the exclusive phytopathogens are not really phytopathogens? Is the study is mainly capturing likely changes in fungal abundance generally? There is no discussion about this.

Response: Given our concerns about the length of the text, we omitted a detailed elaboration on these comparisons in the previous version. It should be noted that the original purpose of dividing phytopathogens into exclusively pathogenic and non-exclusively pathogenic modes was to meet the reviewer's request to test how potential mistakes in functional assignments affect the results. Based on these tests, we observed that both fractions have very similar trends with quantitatively similar results regardless of how the data are split. Thus potentially there is little need to split the PPs into two categories. However, we nonetheless thank the reviewer for this suggestion as it may be important to consider this in more detail in the paper. In this revision, we have now fully expanded on these comparisons, and provided more details and some necessary discussions on the difference between phytopathogens and all fungi. Regarding the broader question of if phytopathogens behave in the same way as all fungi, we cannot fully answer this based on the main dataset, as it would require analysis of all >1M OTUs, which is not feasible. However, previous studies have shown that globally and also regionally, fungal functional groups including phytopathogens display contrasting responses to biotic, edaphic, climatic and spatial conditions (e.g., articles by Tedersoo et al. 2014, 2020, 2022).

1) For relative abundance, we compared the relationships between relative abundance and

latitude of phytopathogens with exclusively phytopathogenic trophic modes to those with non-exclusive trophic modes. In addition, the distribution of phytopathogens is of course different from all fungi from the perspective of relative abundance, since the relative abundance of all fungi in each sample is always 100%, which is independent from latitude (Figure R1 for review only).

Figure R1 Latitudinal distribution of the relative abundance of all fungi (black points), potential phytopathogenic fungi with exclusive trophic modes (plant pathogens only; orange points) and non-exclusive trophic modes (plant pathogen and endophyte and/or saprotrophic fungi; blue points).

See Line 120-124: *However, the phytopathogens with exclusively phytopathogenic trophic modes ($R^2 = 0.005$, $F = 46.6$, Supplementary Fig. 2c) showed a marginally weaker relationship between their relative abundances and latitude than those with non-exclusive trophic modes ($R^2 = 0.006$, $F = 63.6$, Supplementary Fig. 2d).*

2) For diversity, we discussed the difference between phytopathogens and all fungi. Then, we compared results from analyses based on fungi with exclusively phytopathogenic trophic modes to those with non-exclusive trophic modes.

See Line 147-153: *Moreover, the distribution of ppSH diversity is also greatly different from that of total fungi and arbuscular mycorrhizal fungi, whose diversity was demonstrated to peak in tropical areas by multiple previous studies^{1,30-32}. This suggests that fungi with pathotrophic modes may have distinct biogeographical patterns compared to other fungi with different trophic modes (e.g., saprotroph and symbiotroph), as a consequence of differential life-history strategies and community assembly mechanisms³³.*

See Line 160-163: *Similarly, the diversity of phytopathogens with exclusively phytopathogenic trophic modes ($R^2 = 0.054$, $F = 571$, Supplementary Fig. 3g) also showed a marginally weaker relationship with latitude compared with those with non-exclusive trophic modes ($R^2 = 0.066$, $F = 702.6$, Supplementary Fig. 3h).*

3) For community analysis, we compared the Fns value of phytopathogens with other previously published microbial groups including all fungi. (See our response to your Q4 for more details).

See Line 226-231: *The Fns of global potential phytopathogenic fungi observed in the current study (0.12) was lower than those reported for human-associated, bacterial microbiomes (0.23 to 0.99)^{38,39}, and lower than those reported for all fungi (0.63)⁴⁰ and fungi with other trophic modes such as AM fungi in natural and agricultural fields (0.28 to 0.94)^{41,42}. This suggests that the ecological dynamics of phytopathogenic fungi were potentially more host-specific than other microbial groups.*

4) For driving forces, we compared the driving forces acting on phytopathogenic fungi against those acting on all fungi, and then compared the model performances from phytopathogens with exclusively phytopathogenic trophic modes to those with non-exclusive trophic modes.

See Line 255-257: *This is consistent with previous studies on both all fungi and on phytopathogenic fungi, whose global distributions were also mainly determined by climatic factors^{1,13,17}.*

See Line 260-264: *The ppSH richness of phytopathogens with non-exclusive trophic modes was consistently better explained by our random forest models with either solely bioclimatic variables or with the addition of spatial and vegetation variables, both globally and across main land cover types including forest, grassland, cropland, desert, woodland, shrubland, tundra, wetland, and mangrove (Fig. 4a).*

5) For future changes, we compared the difference in the predicted future changes in diversity of phytopathogens with exclusively phytopathogenic trophic modes to those with non-exclusive trophic modes. We replotted Supplementary Figure 8a to make the comparison

clearer.

See Line 298-304: *These results were consistent when data were analysed for potential phytopathogenic fungi with both exclusively phytopathogenic trophic modes and those with non-exclusive trophic modes (Supplementary Fig. 8a). However, the richness of phytopathogens with exclusive trophic modes is expected to have a greater increase than phytopathogens with non-exclusive trophic modes under all future climate scenarios and under all eleven GCMs (Supplementary Fig. 8a).*

Supplementary Fig. 8a The predicted change in diversity of potential phytopathogenic fungi with exclusive trophic modes (plant pathogens only) and non-exclusive trophic modes (plant pathogen and endophyte and/or saprotrophic fungi) under different future climate scenarios. Plot axis labels reflect, shared socioeconomic pathway (SSP); sustainability (SSP126); middle of the road (SSP245); regional rivalry (SSP370); and fossil-fuelled development (SSP585) scenarios. The y-axis shows the relative change in ppSH richness across all samples.

References:

- Tedersoo et al. 2014. Global diversity and geography of soil fungi. *Science* 346: 1078.
- Tedersoo et al. 2020. Regional-scale in-depth analysis of soil fungal diversity reveals strong pH and plant species effects in Northern Europe. *Frontiers in Microbiology* 11:1953.
- Tedersoo et al. 2022. Global patterns in endemism and vulnerability of soil fungi. *Global Change Biology* 28: 6696-6710.

Q3: For the second issue, while I appreciate that data on particular plant hosts would be

another project in and of itself, plant biomass is a poor indicator of plant host range. This caveat needs to be discussed. Despite several of the reviewers bringing this up, it does not seem that the new manuscript even includes the word “host” anywhere. So this caveat is not addressed.

Response: We agree with you that plant biomass is a poor indicator of plant host range, and apologise for not providing a comprehensive discussion of this issue and highlighting the caveats you raised. We have now discussed the issue of host-plant ranges potentially influencing phytopathogen distributions throughout the manuscript. Our revision now included multiple edits (in the analysis and the text) to acknowledge this challenge. First, we integrated the very recently published global plant diversity data (at 1 km resolution) to re-construct our random forest models to provide some insights into the plant factors affecting the distribution of phytopathogens (Fig. 4). Although plant diversity is not a direct indication of host range size, various theories such as Rapoport's rule and the latitudinal gradient in niche breadth suggest that regions with more species are likely to correspond with smaller niche breadths. As such, while we cannot link species, we believe **our vegetation index (indicated by gross primary production and plant diversity)** now can reflect the plant host range to some extent, but as the referee requests, we have also discussed potential caveats of this approach. With this addition, our previous conclusion that climate is more effective than spatial and vegetation variables in predicting ppSH diversity can still be supported. Nevertheless, despite this change, we fully acknowledge that our vegetation index does not fully represent host range (host plant identity, diversity, and abundance), and we still can't link phytopathogen distribution to their particular hosts.

See Line 240-244 & 554-558: *Climatic (indicated by 11 temperature-related and 8 precipitation-related bioclimatic variables), spatial (indicated by longitude, absolute latitude, and standardised principal coordinate of neighbour matrices) and vegetation (indicated by gross primary production and plant diversity) variables were separately or jointly considered in six random forest models.*

See Line 278-284: *Moreover, we acknowledge that our vegetation index only considers plant biomass (gross primary production) and overall plant diversity, and that including*

specific host-plant ranges and associated data (plant species identity, diversity, and abundance) was outside the scope of the current study. These key factors may determine the distribution of phytopathogens, especially at local and regional scales^{44,45}, once the overarching influence of temperature and precipitation on shaping the global distribution of biomes has been accounted for.

See Line 335-338: *We fully acknowledge that there still exists a gap when directly linking the diversity of ppSH to the incidence of plant disease, especially in the context of unknown absolute phytopathogen abundances, and host-plant identities and diversity.*

See Line 385-392: *Furthermore, without global data on the range distribution of plants, directly linking fungal pathogens with actual plant host distributions remains challenging. Our observational data can only provide correlative insights into the distributions of plant pathogens, but a mechanistic understanding of current and future trajectories of pathogen biogeography will require detailed information about host plant ranges. Further research into the distribution of plant species and their niche ranges will be fundamental to facilitating the joint species distribution modelling that is needed to assess these trends in the future.*

See Line 467-472: *The vegetation variables capture gross primary production (GPP) and overall vascular plant diversity. The GPP data used in this study were the annual average GPP data during the last four decades derived from satellite near-infrared reflectance data⁵⁴. The plant diversity data were extracted from the global map of alpha diversity (local species richness, 1 km resolution) for vascular plants built from 170,272 georeferenced local plant assemblages⁵⁵.*

See Line 566-569: *three spatial variables (longitude, absolute latitude, and standardised principal coordinates of neighbour matrices (PCNM)) and two vegetation variables (gross primary production and plant diversity).*

See Legend of Figure 4: *Vegetation reflects gross primary production and plant diversity.*

Q4: Finally, more generally, there are still lots of analyses that are not set in a context to understand what it all means. In particular, it would be helpful if more of the discussion could point out how the patterns differ from all fungi. How are potential phytopathogens different

from all fungi? Or are they the same? (A specific example of a result with no context is the newly added analyses about dissimilarity-overlap curve (Line 203). This is not a well-known metric in the field and would need to be described better. After looking at the citation, I was not surprised that this is the case for environmental habitats. If this metric is used, I would suggest the authors specify their hypothesis and alternatives that might make sense. As a stand alone, it is hard to interpret what this means for the future of phytopathogens, and how this might compare to any other type of fungi.)

Response: We thank the reviewer for pointing this out. Further discussion and comparisons between phytopathogenic fungi and other type of fungi are provided in our revised version (see our response to your Q2 for details). With regard to dissimilarity-overlap curve (DOC) analysis, we also realised that without contextualisation this may be confusing. Given that the intrinsic logic linking DOC analysis and host dependence is both complex and has been fully explained and validated in its initial citation, we decided to simplify this analysis in our manuscript. We explained what this analysis can do, what the index can reflect, and why we choose to use it. Now we explained that the Fns index from the DOC analysis is a parameter assessing the host dependence of a microbiome, and then displayed our results, and compared our results with other previously published reports. We believe the current version is clear and provides a balance between being concise and readable, without overloading the readers with highly specific technical details. However, we are happy to remove this analysis from our paper if you still consider it highly confusing.

See Line 218-237: *To test for host dependence of potential phytopathogenic fungi, the Fns (fraction negative slope) index from dissimilarity-overlap curve (DOC) analysis was employed³⁸. In DOC, a high Fns value indicates that the underlying ecological dynamics of a microbiome are largely host-independent. In contrast, a low Fns value reflects that the ecological dynamics of a microbiome are host-specific³⁸. We observed significant Fns from DOCs across the global potential phytopathogenic fungal dataset ($Fns = 0.12$, $P < 0.001$), and independently across all land cover types (Fns range, 0.002 to 0.428, $P < 0.001$; Fig. 3d), and all habitats (Fns range, 0.002-0.211, $P < 0.001$; Supplementary Fig. 6). The Fns of global potential phytopathogenic fungi observed in the current study*

(0.12) was lower than those reported for human-associated, bacterial microbiomes (0.23 to 0.99)^{38,39}, and lower than those reported for all fungi (0.63)⁴⁰ and fungi with other trophic modes such as AM fungi in natural and agricultural fields (0.28 to 0.94)^{41,42}. This suggests that the ecological dynamics of phytopathogenic fungi were potentially more host-specific than other microbial groups. However, given the small number and scope of these studies, the Fns values for fungi need to be evaluated across more complex ecosystems and at larger scales, as well as via controlled experimental manipulation of factors driving host effects. The relatively high Fns values in cropland (0.35), grassland (0.208), woodland (0.398), shrubland (0.211), and urban (0.428) land cover types, and in soil (0.206) and root (0.211) habitats indicated relatively lower host dependence across these land cover types and/or habitats^{38,41}.

Q5: In sum, while the manuscript presents some interesting analyses, it still is not accompanied with a clear interpretation about what the patterns mean biologically, let alone what the limitations of the study are.

Response: We apologize for failing to satisfactorily address these concerns in the previous revision. We hope that our additional edits will help to clarify the manuscript and address/acknowledge some of the weaknesses raised here.

Minor comments:

Q6: Dust habitat. In the rebuttal letter, it is mentioned that this is atmospheric deposition, but this should be defined in the text, too, as it is used throughout.

Response: Thank you. We have provided the definition for dust in the main text.

See Line 86-88: *In total, our global dataset included 5753 potential phytopathogenic species hypotheses (hereafter ppSHs; ppSHs were generated by 98.5% ITS sequence similarity, which can be treated as potential phytopathogenic fungal species) from 20,312 samples distributed across 11 continents, 11 land cover types (forests, grasslands, croplands, aquatic, deserts, woodlands, shrublands, tundra, wetlands, urban, and mangroves), and 12 habitat types (soils, plant shoots, roots, rhizosphere, deadwood, air, sediment, litter, lichen, freshwater, topsoil, and **dust (atmospheric deposition)**) (Fig. 1).*

Q7: Line 191: “two predominant components” – this is strange wording. Nestedness and turnover are certainly two metrics of community variation, but this suggests somehow that they are dominant (or that we rank different metrics of beta-diversity).

Response: We have changed the word ‘predominant’ to ‘key’.

See Line 206-208: *In community variation, two key components exist – turnover (replacement of species) and nestedness (the extent to which species composition of smaller assemblages is a subset of larger assemblages)³⁷.*

Reviewer #3 (Remarks to the Author):

Q1: The revised version of the manuscript addresses most of the concerns I raised in the previous version, but a few issues remain. There is no doubt that the paper provides a sound comprehensive look at patterns of fungal diversity and relative abundance and concludes that temperature is a major driver of these patterns and that for richness there is a mid-latitude peak in diversity. These are the most sound conclusions. The authors then use random forest models to look at the factors driving individual taxa and MaxEnt style niche models to predict how distributions might change as a function of temperature as a measure of "invasion" potential. Again, think there is little quibble with the data and major patterns, but there remain some issues of interpretation or extrapolation that are of some concern. These have to do with three of the main points from my initial review that the authors address in their response. My comment not their responses are here:

Response: Thank you once again for supporting our study, and we are sorry that some of your concerns were not addressed in our last version. Now we have made further modifications to address these concerns, and hope these modifications are satisfactory. Please see our detailed responses to the points you raised below.

Q2: Point 1. I appreciate the addition of primary productivity (PP) into the models, but this does not really adequately capture the effect of host plant richness and composition, just total production or biomass. I think it is important to acknowledge that there is no real estimate in the model of, for example, how the diversity of plant resources scales with diversity of fungi. PP can be high with either high or low host diversity, so using PP alone doesn't seem to resolve the issue. While it might be true that the range of a particular genus or family is broad, again, I think this is more about the phyletic or functional diversity of hosts within a region. On Line 245, the statement that "vegetation" is of limited importance is grossly overstated. If the authors want to conclude that PP has limited effect that is fine, but that is not the same as saying vegetation doesn't matter. Species composition and local beta diversity of host taxa almost certainly play a role here. The effects of climate might, indeed, act indirectly through plant composition in addition to directly based on abiotic controls. This needs to be acknowledged and also acknowledge that it may complicate effects of future climate.

Response: This is a concern also raised by Reviewer 2. We fully agree with you that GPP does not adequately capture the effect of host plant richness and composition. Here, we integrated the very recently published global plant diversity data (at 1 km resolution) to re-construct our random forest models to provide some insights into the plant factors affecting the distribution of phytopathogens (Fig. 4). Although plant diversity is not a direct indication of host range size, various theories such as Rapoport's rule and the latitudinal gradient in niche breadth suggest that regions with more species are likely to correspond with smaller niche breadths. As such, while we cannot link species, we believe our vegetation index now can reflect the plant host range to some extent. Therefore, in our new version, **vegetation now has two variables, gross primary production and plant diversity**. This vegetation index now reflects the plant host range to some extent, but as both referees request, we have also discussed potential caveats of this approach. With this addition, our previous conclusion that climate is more effective than spatial and vegetation variables in predicting ppSH diversity is still supported. Although we have provided plant diversity, we acknowledge that our vegetation index does not fully represent host range (host plant identity, diversity, and abundance), and we still can't link phytopathogen distribution to their particular hosts. We also acknowledge that climate might indirectly affect the distribution of phytopathogens through regulating the composition of host plant communities, which may provide an additional dimension to the effects of future climate on distribution of phytopathogens.

See Line 240-244 & 554-558: *Climatic (indicated by 11 temperature-related and 8 precipitation-related bioclimatic variables), spatial (indicated by longitude, absolute latitude, and standardised principal coordinate of neighbour matrices) and vegetation (indicated by gross primary production and plant diversity) variables were separately or jointly considered in six random forest models.*

See Line 274-284: *However, climate change might indirectly affect the distribution of phytopathogens through modifying the composition of host-plant communities, which may provide an additional dimension to the effects of future climate on the distribution of phytopathogens. Moreover, we acknowledge that our vegetation index only considers plant biomass (gross primary production) and overall plant diversity, and that including*

specific host-plant ranges and associated data (plant species identity, diversity, and abundance) was outside the scope of the current study. These key factors may determine the distribution of phytopathogens, especially at local and regional scales^{44,45}, once the overarching influence of temperature and precipitation on shaping the global distribution of biomes has been accounted for.

See Line 335-338: *We fully acknowledge that there still exists a gap when directly linking the diversity of ppSH to the incidence of plant disease, especially in the context of unknown absolute phytopathogen abundances, and host-plant identities and diversity.*

See Line 385-392: *Furthermore, without global data on the range distribution of plants, directly linking fungal pathogens with actual plant host distributions remains challenging. Our observational data can only provide correlative insights into the distributions of plant pathogens, but a mechanistic understanding of current and future trajectories of pathogen biogeography will require detailed information about host plant ranges. Further research into the distribution of plant species and their niche ranges will be fundamental to facilitating the joint species distribution modelling that is needed to assess these trends in the future.*

See Line 467-472: *The vegetation variables capture gross primary production (GPP) and overall vascular plant diversity. The GPP data used in this study were the annual average GPP data during the last four decades derived from satellite near-infrared reflectance data⁵⁴. The plant diversity data were extracted from the global map of alpha diversity (local species richness, 1 km resolution) for vascular plants built from 170,272 georeferenced local plant assemblages⁵⁵.*

See Line 566-569: *three spatial variables (longitude, absolute latitude, and standardised principal coordinates of neighbour matrices (PCNM)) and two vegetation variables (gross primary production and plant diversity).*

See Legend of Figure 4: *Vegetation reflects gross primary production and plant diversity.*

Q3: Point 2. I agree that diversity and relative abundance are useful, but it is important to recognize that the ecological effect of each of these is very uncertain. There is considerable literature that shows diversity affects ecosystem processes, but inferring the direction of those

effects with respect to a particular process like disease is a considerable leap in logic that needs better support or more explanation about the potential range of outcomes of a change in diversity. For example high diversity could imply more disease with a broader range of potential hosts affected or lower disease risk due to dilution and lower density of fungal propagules of any one ppSH.

Response: We agree. Although diversity-function relationships have been documented in many ecosystems, a direct link between pathogen diversity and plant disease may be illogical. However, as this part (the potential link between ppSH diversity and plant disease) can strengthen the significance of the current study, we still wish to include the content, which is supported from research data rather than whimsical speculation. Based on this, we have now acknowledged the gap between ppSH diversity and plant disease incidence in our revised manuscript.

See Line 335-346: *We fully acknowledge that there still exists a gap when directly linking the diversity of ppSH to the incidence of plant disease, especially in the context of unknown absolute phytopathogen abundances, and host-plant identities and diversity. For example, high phytopathogen diversity could imply more disease with a broader range of potential hosts affected, but may also lead to lower disease risk due to dilution effects and lower densities of host-specific fungal propagules. Moreover, diversity does not fully scale with absolute abundance, and integrating absolute abundance data of all fungi/phytopathogenic fungi would increase the predictive accuracy of plant disease incidence. Therefore, further experimentation is required to fully examine the mechanisms underpinning the diversity-disease severity relationship, and the contrasting responses of potential phytopathogenic fungal diversity from different land cover types and habitats to climate change.*

Q4: Point 3. I appreciate the resampling approach, however the issue with the approach described is that resampling from all regions that have > 300 samples can be confounded by the spatial distribution of those samples... if some regions have far more than 300 samples then they likely sample a broader range of habitat types and so the resampling approach will (by chance) tend to produce greater diversity in those areas. If there is no relationship

between sampling intensity and latitude, then just say that and you can move on. If there is such a relationship, then I think this needs to be mentioned as an important caveat along with the resampling approach mentioned above. I am not saying it negates the conclusions, just that it needs to be acknowledged. Specifically, I would like to see mentioned the fact that the sampling appears to be biased at the 26-32 degree latitude to humid areas of E and SE Asia and that the unequal longitudinal distribution of latitudinal samples might confound interpretation. Because the authors included longitude a variable in these analyses, it would seem that these effects do not negate the latitudinal patterns (though incorporating longitude is often tricky because of the circular nature of the globe... the authors say they used absolute longitude... does that mean that they used absolute value (0 to 180) or absolute (+180 to -180). If the latter than this seems the best one can do, but if the former, then this needs to be redone since it is hard to imagine why +90 and -90 would be expected to have the same fungal diversity values.

Response: Thank you for supporting our resampling approach.

(1) We tested the sampling density alongside the latitude, finding that sampling density was highest at $\sim 45^\circ$, which does not align with the region of peaked ppSH diversity ($\sim 26-32^\circ$) (Figure R2 for review only). This suggest that our conclusion is not biased by an unbalance sampling density. However, after careful consideration, we find the point you raised is exactly correct, and it is an important statistical detail that is easily overlooked. Therefore, we now acknowledged this in our revised version after the section of resampling approach.

See Line 171-175: *Although our resampling approach can, to a great extent, avoid the bias from unbalanced sampling, it's worth noting that resampling from highly dense sampling regions, such as humid areas of East and Southeast Asia in this study, could also potentially produce greater diversity in those areas.*

(2) You raised the question on longitude. Sorry that our descriptions were confusing (our initial description is 'indicated by absolute latitude, longitude, and standardized principal coordinate of neighbour matrices'). The longitude value we used is the raw value (-180 to 180). We have rewritten this sentence to avoid confusion.

See Line 240-244: *Climatic (indicated by 11 temperature-related and 8 precipitation-related*

bioclimatic variables), spatial (indicated by longitude, absolute latitude, and standardised principal coordinate of neighbour matrices) and vegetation (indicated by gross primary production and plant diversity) variables were separately or jointly considered in six random forest models.

Figure R2 Sampling density alongside the latitude.

Two additional concerns:

Q5: I still do not understand what universal dynamics means. Perhaps this is something known in the fungal ecology world, but as I have never heard this term and it needs a brief explanation in the text (not just in supplemental) if it is to be used here.

Response: We are sorry for the confusion and the lack of clarity in our previous version. Given that the intrinsic logic linking DOC analysis and host dependence is both complex and has been fully explained and validated in its initial citation, we decided to simplify this analysis in our manuscript. We explained what this analysis can do, what the index can reflect, and why we choose to use it. Now we explained that the Fns index from the DOC analysis is a parameter assessing the host dependence of a microbiome, and then displayed our results, and compared our results with other previously published reports. We believe the current version is clear and provides a balance between being concise and readable, without overloading the readers with highly specific technical details. However, if the reviewer suggests, we are happy to remove this analysis from our paper to improve clarity if needed.

See Line 218-237: *To test for host dependence of potential phytopathogenic fungi, the Fns*

(fraction negative slope) index from dissimilarity-overlap curve (DOC) analysis was employed³⁸. In DOC, a high Fns value indicates that the underlying ecological dynamics of a microbiome are largely host-independent. In contrast, a low Fns value reflects that the ecological dynamics of a microbiome are host-specific³⁸. We observed significant Fns from DOCs across the global potential phytopathogenic fungal dataset (Fns = 0.12, $P < 0.001$), and independently across all land cover types (Fns range, 0.002 to 0.428, $P < 0.001$; Fig. 3d), and all habitats (Fns range, 0.002-0.211, $P < 0.001$; Supplementary Fig. 6). The Fns of global potential phytopathogenic fungi observed in the current study (0.12) was lower than those reported for human-associated, bacterial microbiomes (0.23 to 0.99)^{38,39}, and lower than those reported for all fungi (0.63)⁴⁰ and fungi with other trophic modes such as AM fungi in natural and agricultural fields (0.28 to 0.94)^{41,42}. This suggests that the ecological dynamics of phytopathogenic fungi were potentially more host-specific than other microbial groups. However, given the small number and scope of these studies, the Fns values for fungi need to be evaluated across more complex ecosystems and at larger scales, as well as via controlled experimental manipulation of factors driving host effects. The relatively high Fns values in cropland (0.35), grassland (0.208), woodland (0.398), shrubland (0.211), and urban (0.428) land cover types, and in soil (0.206) and root (0.211) habitats indicated relatively lower host dependence across these land cover types and/or habitats^{38,41}.

Q6: I appreciate that current analyses do not allow quantification of absolute abundances of ppSHs and that the present analyses still have value with richness, diversity and relative abundances. That said, I think the authors should acknowledge how knowing patterns of absolute abundance might alter the conclusions of the present manuscript. Total fungal abundance could be assessed and presumably this would scale largely with humidity, at least to a first order approximation. If that seems reasonable, how would knowing that alter the conclusions of the authors regarding the effects of changing climate on fungal effects on plants.

Response: We agree. The absolute abundance of fungi or phytopathogenic fungi is of great importance to predict plant disease incidence. We acknowledged this in our new revision,

after displaying the potential links between phytopathogen diversity and plant disease incidence.

See Line 335-346: *We fully acknowledge that there still exists a gap when directly linking the diversity of ppSH to the incidence of plant disease, especially in the context of unknown absolute phytopathogen abundances, and host-plant identities and diversity. For example, high phytopathogen diversity could imply more disease with a broader range of potential hosts affected, but may also lead to lower disease risk due to dilution effects and lower densities of host-specific fungal propagules. Moreover, diversity does not fully scale with absolute abundance, and integrating absolute abundance data of all fungi/phytopathogenic fungi would increase the predictive accuracy of plant disease incidence. Therefore, further experimentation is required to fully examine the mechanisms underpinning the diversity-disease severity relationship, and the contrasting responses of potential phytopathogenic fungal diversity from different land cover types and habitats to climate change.*

REVIEWER COMMENTS

Reviewer #2 (Remarks to the Author):

I appreciate that the authors have tried to clarify the text and have added in the caveats as requested.

Reviewer #3 (Remarks to the Author):

Overall, the authors now ACKNOWLEDGE the concerns and criticisms that I related in my previous reviews regarding the effect of host plant range, oversampling in humid areas of east Asia and the distinction between ppsh and other fungi. I think these concerns are generally appropriately acknowledged, and the authors have done the best that is possible given the existing data to address the concerns substantially (e.g., by including plant diversity data) and acknowledge their limitations. That said, they don't really evaluate what these acknowledgments and caveats mean with respect to the authors' ability to address the main questions posed in the paper.

For example, the introduction makes it clear that a driving motivation of the study is (L55) "the effective management of fungal pathogens, both now and in the future, requires a comprehensive understanding of the ecological mechanisms driving the diversity and distribution of pathogenic fungi..." The latitudinal pattern explains less than 1% in the variation of ppsh diversity with latitude, and as previously discussed it is not clear what relative abundance means for disease management. As I said in my previous review, "there is little quibble with the data and major patterns, but there remain some issues of interpretation or extrapolation that are of some concern." I think the authors have done the best they possibly can given the limitations of the data to address the issues I raise. At this point, I think its up to the editor to decide whether the acknowledgments of limitations undermine the main goals of the ms to such an extent that the paper needs to be recast as a more basic description of patterns with limited implications for understanding management of plant disease.

Some specific comments on the most recent version.

L66: "Given the tightly coupled relationships between climate and pathogen diversity(ref18)..." . This seems to indicate that one of the main findings of the current manuscript was previously established in the literature. It would be good for the authors to clarify what the new contribution of this work is in the space of pathogen and climate variables.

L221 (and more generally the section on universal dynamics). The authors explain these approaches more clearly in the methods now, which I appreciate. I still do not understand the logic for how the deviation from the DOC tells you something about host specificity, but assuming that is correct, the authors conclude here that (Lin 230): "This suggests that the ecological dynamics of phytopathogenic fungi were potentially more host-specific than other microbial groups." I found this worrisome given the conclusions elsewhere in the manuscript that climate was the main driver and vegetation factors were of limited importance (e.g, paragraph starting on L 248). If this inference from the DOC and Fns are correct, then it implies that the authors' attempts at including host variables in the random forest models remains wholly inadequate. I see this as a major concern regarding the interpretation of the data as useful for projecting concerns about plant disease in the present and future world. Again, the data and basic patterns are sound, but I remain concerned that the interpretation of the data for management-- a stated rationale for the study in the introduction-- is problematic or limited.

L336. The authors now acknowledge the issues with interpreting ppsh diversity in terms of disease risk here, stating "We fully acknowledge that there still exists a gap when directly linking the diversity of ppSH to the incidence of plant disease, especially in the context of unknown absolute phytopathogen abundances, and host-plant identities and diversity." Going on to elaborate on how diversity could increase or decrease disease risk-- given these caveats (which I agree with), is it safe to say that patterns of pathogen diversity alone tell us very little about disease risk now and in the future?

Minor comments:

L410. Can you briefly explain what designation of pp as probable and highly probable mean? Is there a percent probability associated with these designations?

L434: Sampling is reported from 11 continents... last I checked where were only 7.

L462: Not sure what journal style is here, but citing a website is always problematic since there is no guarantee the information will be there when a readers goes to look. Is there a stable location for this information that the authors can cite?

Q4 Authors response: Longitude is coded as -180 to +180... this means that sampling right next to each other at -180 vs +180 would indicate extreme difference in longitude despite being adjacent to each other. That said, since this is a land-based study and the - and + 180 longitudes are in the Pacific Ocean, this probably has little influence on the analysis.

Alternatively, the authors could bin longitude and analyze as a random unordered factor. I do wonder if they did that whether the East Asian longitude bin might have a strong effect?

Point-by-point response to reviewers' comments

REVIEWER COMMENTS

Reviewer #2 (Remarks to the Author):

I appreciate that the authors have tried to clarify the text and have added in the caveats as requested.

Response: Thank you for being satisfied with our revisions.

Reviewer #3 (Remarks to the Author):

Q1: Overall, the authors now ACKNOWLEDGE the concerns and criticisms that I related in my previous reviews regarding the effect of host plant range, oversampling in humid areas of east Asia and the distinction between ppsh and other fungi. I think these concerns are generally appropriately acknowledged, and the authors have done the best that is possible given the existing data to address the concerns substantially (e.g., by including plant diversity data) and acknowledge their limitations. That said, they don't really evaluate what these acknowledgments and caveats mean with respect to the authors' ability to address the main questions posed in the paper.

For example, the introduction makes it clear that a driving motivation of the study is (L55) "the effective management of fungal pathogens, both now and in the future, requires a comprehensive understanding of the ecological mechanisms driving the diversity and distribution of pathogenic fungi..." The latitudinal pattern explains less than 1% in the variation of ppsh diversity with latitude, and as previously discussed it is not clear what relative abundance means for disease management. As I said in my previous review, "there is little quibble with the data and major patterns, but there remain some issues of interpretation or extrapolation that are of some concern."

I think the authors have done the best they possibly can given the limitations of the data to address the issues I raise. At this point, I think its up to the editor to decide whether the acknowledgments of limitations undermine the main goals of the ms to such an extent that the paper needs to be recast as a more basic description of patterns with limited implications for understanding management of plant disease.

Response: We thank you once again for raising these points to us. We think what concerned you

most in our revised manuscript was the descriptions related to 1) disease management and 2) the real evaluation on limitations. Following yours and the editor's suggestions, we now made some further modifications and adjustments to avoid the misinterpretation. Please see below for our detailed modifications on these issues.

1) The issue on plant disease management

Our initial driving motivation conducting this study is to understand the global distribution, regulating factor and future potential changes of phytopathogenic fungi, and we think this will provide foundational knowledge that will contribute to plant disease management. However, our initial descriptions may be somewhat confusing, making readers wonder if the real motivation and ultimate goal of this study is disease management. To avoid this confusion, we have further clarified our motivation in the first paragraph of Introduction, by removing the statement on plant disease management.

See Line 55-57: Therefore, ascertaining the distribution and the environmental attributes that structure phytopathogenic fungal communities across the globe was recently considered to be a priority research direction⁸.

However, we think the statement at the end of paragraph #2 in Introduction, '*Considering the magnitude of global climate change, it is imperative to determine how a changing climate affects the distribution of phytopathogenic fungi, and potentially to use this new knowledge to inform policies to control the emergence of future plant diseases and maintain ecosystem functions and services*' is safe enough, which will not be misleading, and we therefore think this should be maintained.

2) The real evaluation on limitations

We carefully checked the limitations we acknowledged, and we think most are appropriate. However, the discussions on pathogen diversity-plant disease relationship may not be enough, and further discussions on this issue were provided. Here, we explicitly stated that the relationship between ppSH diversity and CPD emergence only represents a statistical attempt in cropland, and caution is required when extrapolating the positive ppSH-CPD relationship in the current study to natural ecosystems. The limitations were fully acknowledged and discussed using an extended paragraph including covering how diversity could increase or decrease disease risk, and we

believe this would not lead to misinterpretation or premature application of the findings by readers. Please also see our response to Q4.

See Line 338-348: It should be noted that direct links between the diversity and relative abundances of ppSH and the incidence of plant disease are not yet established. For example, high phytopathogen diversity could imply greater disease incidence with a broader range of potential hosts affected, but may also lead to lower disease risk due to dilution effects and lower densities of host-specific fungal propagules. Furthermore, pathogen diversity may be a key regulator maintaining the plant diversity by relatively stronger suppression of dominant species, hence preventing competitive exclusion^{46, 47}. Therefore, the observed positive correlation between ppSH diversity and CPD emergence in croplands may not hold in other land cover types, and caution is required when extrapolating the ppSH-disease relationships of anthropogenic habitats to natural ecosystems.

See Line 348-355: Moreover, diversity and relative abundances may be unrelated to absolute abundances, which could not be deduced from our datasets. For example, high phytopathogen diversity with low absolute abundance may not bring greater disease risk compared with low phytopathogen diversity with high absolute abundances. Therefore, further experimentation is required to fully examine the mechanisms underpinning the diversity-disease severity relationship, and the contrasting responses of potential phytopathogenic fungal diversity from different land cover types and habitats to climate change.

Some specific comments on the most recent version.

Q2: L66: "Given the tightly coupled relationships between climate and pathogen diversity(ref18)..." . This seems to indicate that one of the main findings of the current manuscript was previously established in the literature. It would be good for the authors to clarify what the new contribution of this work is in the space of pathogen and climate variables.

Response: We apologize for the incorrect description and citation here. The original statement in this citation is that *climate warming can increase pathogen development and survival rates*. They didn't assess the relationship between climate and pathogen diversity. We have corrected the description here. Our work is the first large-scale study investigating the relationship

between climate and phytopathogen diversity.

See Line 66-68: Given the tightly coupled relationships between **climate and pathogen development**¹⁸, climate change has increased the prevalence and severity of some human, animal and plant diseases¹⁹.

Q3: L221 (and more generally the section on universal dynamics). The authors explain these approaches more clearly in the methods now, which I appreciate. I still do not understand the logic for how the deviation from the DOC tells you something about host specificity, but assuming that is correct, the authors conclude here that (Lin 230): "This suggests that the ecological dynamics of phytopathogenic fungi were potentially more host-specific than other microbial groups." I found this worrisome given the conclusions elsewhere in the manuscript that climate was the main driver and vegetation factors were of limited importance (e.g, paragraph starting on L 248). If this inference from the DOC and Fns are correct, then it implies that the authors' attempts at including host variables in the random forest models remains wholly inadequate. I see this as a major concern regarding the interpretation of the data as useful for projecting concerns about plant disease in the present and future world. Again, the data and basic patterns are sound, but I remain concerned that the interpretation of the data for management-- a stated rationale for the study in the introduction-- is problematic or limited.

Response: The DOC analysis tested the host dependence of a microbiome based on statistical methods, without considering the real host data. Our results showed that the ppSHs were more influenced by climate variables than by host plants, although ppSHs were more host-dependent than other microbial groups. Let's take a look at the hypothetical Figure R1 for an example. Although ppSHs may be more influenced by host than other microbes, ppSHs were still more affected by climate variables. So, the outputs of DOC analysis and random forest analysis are not contradictory or conflicting, and the future projections based on climate data should not be a concern in the current study. Moreover, following your previous suggestion, we have changed the description that '*vegetation played a limited role*' into '*vegetation played a relatively small role globally compared to bioclimatic variables in determining both the diversity and relative abundance of ppSHs*', which is more accurate and logically correct. Furthermore, we removed the statement on disease management in the Introduction to avoid overselling our story. Lastly, the

DOC analysis can be removed from our story if you strongly disagree this analysis and explicitly recommend the deletion.

Figure R1 for review | A hypothetical display showing the relative influence of climate and host on the distribution of phytopathogenic fungi and other microbial groups.

Q4: L336. The authors now acknowledge the issues with interpreting ppsh diversity in terms of disease risk here, stating "We fully acknowledge that there still exists a gap when directly linking the diversity of ppSH to the incidence of plant disease, especially in the context of unknown absolute phytopathogen abundances, and host-plant identities and diversity." Going on to elaborate on how diversity could increase or decrease disease risk-- given these caveats (which I agree with), is it safe to say that patterns of pathogen diversity alone tell us very little about disease risk now and in the future?

Response: Our data showed a positive diversity-disease correlation (Pearson's $r = 0.405$, $P = 0.002$, Supplementary Fig. 8b), but we appreciate it is inappropriate to say that high pathogen diversity alone promotes disease risk. Arguably, the safest (or at least most cautious) way forward is to remove the diversity-disease relationship. However, we still think the statistical diversity-disease relationship we provide here is a valid statistical result. Although there remain some limitations when linking them together, the diversity-disease relationship in the current study may guide future researches. In our revised version, the limitations are fully acknowledged and further discussed in an extended paragraph, including covering how diversity could increase or decrease disease risk, and we believe this would not lead to misinterpretation or premature application of the findings by readers.

See Line 338-348: It should be noted that direct links between the diversity and relative abundances of ppSH and the incidence of plant disease are not yet established. For example, high phytopathogen diversity could imply greater disease incidence with a broader range of potential hosts affected, but may also lead to lower disease risk due to dilution effects and lower densities of host-specific fungal propagules. Furthermore, pathogen diversity may be a key regulator maintaining the plant diversity by relatively stronger suppression of dominant species, hence preventing competitive exclusion^{46, 47}. Therefore, the observed positive correlation between ppSH diversity and CPD emergence in croplands may not hold in other land cover types, and caution is required when extrapolating the ppSH-disease relationships of anthropogenic habitats to natural ecosystems.

See Line 348-355: Moreover, diversity and relative abundances may be unrelated to absolute abundances, which could not be deduced from our datasets. For example, high phytopathogen diversity with low absolute abundance may not bring greater disease risk compared with low phytopathogen diversity with high absolute abundances. Therefore, further experimentation is required to fully examine the mechanisms underpinning the diversity-disease severity relationship, and the contrasting responses of potential phytopathogenic fungal diversity from different land cover types and habitats to climate change.

Minor comments:

Q5: L410. Can you briefly explain what designation of pp as probable and highly probable mean?

Is there a percent probability associated with these designations?

Response: The initial publication of FUNGuild says ‘For all database entries a confidence ranking (“highly probable”, “probable”, and “possible”) has been included, reflecting the likelihood that a taxon belongs to a given guild. Whenever possible, confidence assignments were based on assessments given in primary research literature’. Therefore, the confidence ranking (probable and highly probable) are qualitative and based on expert knowledge, and therefore, a quantitative percent probability doesn’t exist.

Q6: L434: Sampling is reported from 11 continents... last I checked where were only 7.

Response: That depends how to define ‘continent’. We agree with you that there are only 7 continents on Earth, from a traditional geographical perspective. The GlobalFungi database divided the Earth into 11 continents, after including the 4 oceans. Given that our data were mainly derived from GlobalFungi, we didn’t want to change their data structure since this may weaken the reproducibility of this study. If we must use 7 continents, then the samples collected from islands in oceans should be removed, because it’s not appropriate to group them into the 7 continents.

Q7: L462: Not sure what journal style is here, but citing a website is always problematic since there is no guarantee the information will be there when a readers goes to look. Is there a stable location for this information that the authors can cite?

Response: Thank you for pointing this out. Yes, these scenarios have been described in a permanent publication, and we have corrected this here.

See Line 467-470: Monthly values of minimum temperature, maximum temperature, and precipitation were processed for four Shared Socio-economic Pathways (SSP): 126, 245, 370 and 585 (SSP126: sustainability; SSP245: middle of the road; SSP370: regional rivalry; SSP585: fossil-fuelled development)⁵⁷.

Reference 57: O’Neill, B.C. et al. A new scenario framework for climate change research: the concept of shared socioeconomic pathways. *Climatic Change* **122**, 387-400 (2014).

Q8: Q4 Authors response: Longitude is coded as -180 to +180... this means that sampling right next to each other at -180 vs +180 would indicate extreme difference in longitude despite being adjacent to each other. That said, since this is a land-based study and the - and + 180 longitudes are in the Pacific Ocean, this probably has little influence on the analysis. Alternatively, the authors could bin longitude and analyze as a random unordered factor. I do wonder if they did that whether the East Asian longitude bin might have a strong effect?

Response: This is an interesting suggestion. Following your suggestion, we tried to re-conduct the random forest analysis by binning samples according to the longitude information at a 10-degree resolution. First, the ordered binning was conducted, and the results showed that the explained

variation in ppSH richness by model #2 (sole spatial variables) declined compared to the model using raw longitude value (**Figure R2** for review). Second, the unordered binning was conducted, and the results showed that the explained variation further declined. We even tried to conduct the random forest by directly removing the variable longitude, and the explained variation sharply declined, suggesting that longitude should not be removed from this analysis. We're not sure if doing the binning procedure reflects the real situation, or underestimates the significance of spatial variables in explaining the ppSH richness. But given that either binning or not binning did not influence our conclusion, we still tend to use the raw longitude values, because longitude binning is unfamiliar, and conducting the binning procedure would be highly confusing to most readers, even we provide many sentences and some supplementary figures to explain how longitude binning is done. Moreover, we have never seen longitude binning in previous studies, and we therefore, are not sure if this is really the optimal solution. Finally, we also plotted a longitudinal distribution of ppSH richness to satisfy your curiosity on the East Asian (longitude range: about 90-150, but Australia is also included; **Figure R3** for review).

Figure R2 for review | The explained variation in ppSH richness by space model.

Figure R3 for review | The longitudinal distribution of ppSH richness.